# Certified Robustness to Data Poisoning in Gradient-Based Training

**Philip Sosnin**                                                                 *p.sosnin23@imperial.ac.uk*
*Department of Computing, Imperial College London, United Kingdom*

**Mark N. Müller**                                                                 *mark.mueller@inf.ethz.ch*
*Department of Computer Science, ETH Zurich, Switzerland*
*LogicStar.ai, Switzerland*

**Maximilian Baader**                                                               *mbaader@inf.ethz.ch*
*Department of Computer Science, ETH Zurich, Switzerland*

**Calvin Tsay**[*]                                                                   *c.tsay@imperial.ac.uk*
*Department of Computing, Imperial College London, United Kingdom*

**Matthew Wicker**[*]                                                               *m.wicker@imperial.ac.uk*
*Department of Computing, Imperial College London, United Kingdom*
*The Alan Turing Institute, United Kingdom*

**Reviewed on OpenReview:** *https://openreview.net/forum?id=9WHifn9ZVX*

## Abstract

Modern machine learning pipelines leverage large amounts of public data, making it infeasible to guarantee data quality and leaving models open to poisoning and backdoor attacks. Provably bounding the behavior of learning algorithms under such attacks remains an open problem. In this work, we address this challenge by developing the first framework providing provable guarantees on the behavior of models trained with potentially manipulated data without modifying the model or learning algorithm. In particular, our framework certifies robustness against untargeted and targeted poisoning, as well as backdoor attacks, for bounded and unbounded manipulations of the training inputs and labels. Our method leverages convex relaxations to over-approximate the set of all possible parameter updates for a given poisoning threat model, allowing us to bound the set of all reachable parameters for any gradient-based learning algorithm. Given this set of parameters, we provide bounds on worst-case behavior, including model performance and backdoor success rate. We demonstrate our approach on multiple real-world datasets from applications including energy consumption, medical imaging, and autonomous driving.

## 1 Introduction

To achieve state-of-the-art performance, modern machine learning pipelines involve pre-training on massive, uncurated datasets; subsequently, models are fine-tuned with task-specific data to maximize downstream performance (Han et al., 2021). Unfortunately, the datasets used in both steps are potentially untrustworthy and of such scale that rigorous quality checks become impractical.

Yet, adversarial manipulation, i.e., *poisoning attacks*, affecting even a small proportion of data used for either pre-training or fine-tuning can lead to catastrophic model failures (Carlini et al., 2023). For instance, Yang et al. (2017) show how popular recommender systems on sites such as YouTube, Ebay, and Yelp can be easily manipulated by poisoning. Likewise, Zhu et al. (2019) show that poisoning even 1% of training data

---

[*]Equal contribution

can lead models to misclassify targeted examples, and Han et al. (2022) use poisoning to selectively trigger backdoor vulnerabilities in lane detection systems to force critical errors.

Despite the gravity of the failure modes induced by poisoning attacks, counter-measures are generally attack-specific and only defend against known attack methods (Tian et al., 2022). The result of attack-specific defenses is an effective "arms race" between attackers trying to circumvent the latest defenses and counter-measures being developed for the new attacks. In effect, even best practices, i.e., using the latest defenses, provide no guarantees of protection against poisoning attacks. To date, relatively few approaches have sought provable guarantees against poisoning attacks. These methods are often limited in scope, e.g., applying only to linear models (Rosenfeld et al., 2020; Steinhardt et al., 2017), only providing approximate guarantees (Xie et al., 2022), or only working for some limited poisoning settings (Rosenfeld et al., 2020). Other approaches partition datasets into hundreds or thousands of disjoint shards and then aggregate predictions such that the effects of poisoning are provably limited (Levine & Feizi, 2020; Wang et al., 2022). In contrast, our goal in this work is not to produce a robust learning algorithm, but to efficiently analyze the sensitivity of (un-modified) algorithms. We provide an extensive discussion of related works in Appendix A.

**This Work: General Certificates of Poisoning Robustness.** We present an approach for computing sound certificates of robustness to poisoning attacks for any model trained with first-order optimization methods, e.g., stochastic gradient descent or Adam (Kingma & Ba, 2014). Unlike prior works we highlight that our guarantees of robustness apply to general poisoning adversaries, i.e., adversaries able to arbitrarily and adaptively modify a bounded number of training data and labels. Our analysis applies to a given model and training algorithm (without need for modification) and therefore can be used to analyze and investigate the poisoning robustness of various proposed defenses and benchmarks. The proposed strategy begins by treating various poisoning attacks as constraints over an adversary's perturbation 'budget' in input and label spaces. Following the comprehensive taxonomy by Tian et al. (2022), we view the objective of each poisoning attack as an optimization problem. Though our framework is general we explicitly consider three objectives: (i) untargeted attacks: reducing model performance to cause denial-of-service, (ii) targeted attacks: compromising model performance on certain types of inputs, and (iii) backdoor attacks: leaving the model performance stable, but introducing a trigger pattern that causes errors at deployment time. Our approach then leverages convex relaxations of both the training problem and the constraint sets defining the threat model to compute *a sound (but potentially incomplete) certificate that bounds the impact of the poisoning attack.* In summary, this paper makes the following key contributions:

- A framework for soundly bounding the reachable parameter space of gradient-trained models under poisoning attacks.

- An instantiation of our framework based on bound-propagation, including a novel extension of the CROWN algorithm to the interval-parameter setting.

- Based on the above, a series of formal proofs that allow us to bound the effect of poisoning attacks with respect to arbitrary attack goals.

- An extensive empirical evaluation demonstrating the effectiveness of our approach.

## 2 Background: Poisoning Attacks

This section describes the capabilities of the poisoning attack adversaries that we seek to certify against. We consider two distinct threat models of bounded and unbounded poisoning attacks, which we define below. Typical threat models additionally specify the adversary's system knowledge; however, we aim to upper bound a worst-case adversary, so assume unrestricted access to all training information including model architecture and initialization, data, data ordering, hyper-parameters, etc.

**Notation.** We denote a machine learning model as a parametric function $f$ with parameters $\theta$, feature space $x \in \mathbb{R}^{n_{\text{in}}}$, and label space $y \in \mathcal{Y}$. The label space $\mathcal{Y}$ may be discrete (e.g. classification) or continuous (e.g. regression). We operate in the supervised learning setting with a dataset $\mathcal{D} = \{(x^{(i)}, y^{(i)})\}_{i=1}^{N}$. We denote the parameter initialization $\theta'$ and a training algorithm $M$ as $\theta = M(f, \theta', \mathcal{D})$, i.e., given a model,

---

**Algorithm 1** ABSTRACT GRADIENT TRAINING FOR COMPUTING VALID PARAMETER-SPACE BOUNDS

---

1: **input:** $f$ - model, $\theta'$ - init. params., $D$ - dataset, $E$ - epochs, $\alpha$ - learning rate, $\mathcal{T}$ - allowable dataset perturbations (poisoning adversary), $\kappa$ - optional clipping parameter.

2: **output:** $\theta$ - nominal SGD parameter, $[\theta_L, \theta_U]$ - valid parameter space bound on the influence of $\mathcal{T}$

3: $\theta \leftarrow \theta'$; $[\theta_L, \theta_U] \leftarrow [\theta', \theta']$             // *Initialize nominal parameter and interval bounds.*

4: **for** $E$-many epochs **do**

5:    **for** each batch $\mathcal{B} \subset D$ **do**

6:       $\Delta\theta \leftarrow \dfrac{1}{|\mathcal{B}|} \sum_{(x,y)\in\mathcal{B}} \text{Clip}_\kappa \left[ \nabla_\theta \mathcal{L} \left( f^\theta(x), y \right) \right]$     // *Compute nominal SGD parameter update (clipping optional).*

7:       $\theta \leftarrow \theta - \alpha\Delta\theta$                 // *Update the nominal parameter.*

8:       $\Delta\Theta \leftarrow \left\{ \dfrac{1}{|\widetilde{\mathcal{B}}|} \sum_{(\tilde{x},\tilde{y})\in\widetilde{\mathcal{B}}} \text{Clip}_\kappa \left[ \nabla_{\tilde{\theta}} \mathcal{L} \left( f^{\tilde\theta}(\tilde{x}), \tilde{y} \right) \right] \mid \widetilde{\mathcal{B}} \in \mathcal{T}(\mathcal{B}), \tilde{\theta} \in [\theta_L, \theta_U] \right\}$     // *Define the set of descent directions under $\mathcal{T}$.*

9:       Compute $\Delta\theta_L$, $\Delta\theta_U$ s.t. $\Delta\theta_L \leq \Delta\theta \leq \Delta\theta_U$    $\forall \Delta\theta \in \Delta\Theta$     // *Bound possible descent directions under $\mathcal{T}$.*

10:      $\theta_L \leftarrow \theta_L - \alpha\Delta\theta_U$;    $\theta_U \leftarrow \theta_U - \alpha\Delta\theta_L$     // *Update parameter interval reachable under $\mathcal{T}$.*

11:    **end for**

12: **end for**

13: **return** $\theta, [\theta_L, \theta_U]$

---

initialization, and data, the training function $M$ returns a "trained" parameterization $\theta$. Finally, we assume the loss function is computed element-wise from the dataset, denoted as $\mathcal{L}(f(x^{(i)}), y^{(i)})$.

**Bounded Poisoning Attacks.** In a bounded attack setting, an adversary is permitted to perturb a subset of the training data in both the feature and label spaces, with the perturbation constrained by a given norm. Specifically, the adversary may select up to $n$ data-points, modifying their features by at most $\epsilon$ in the $\ell_p$-norm and their labels by at most $\nu$ in the $\ell_q$-norm. In a classification setting, label-flipping attacks can be considered under this attack model by setting $\nu = 1, q = 0$. Given a dataset $\mathcal{D}$ and an adversary $\langle n, \epsilon, p, \nu, q \rangle$, we denote the set of potentially poisoned datasets as $\mathcal{T}^{\langle n, \epsilon, p, \nu, q \rangle}(\mathcal{D})$.

**Unbounded Poisoning Attacks.** It may not always be realistic to assume that the effect of a poisoning adversary is bounded. A more powerful adversary may be able to inject arbitrary data-points into the training data set, for example by exploiting the collection of user data. In this "unbounded" attack setting, we use $\mathcal{T}^n$ denote the set of all possible datasets derived from a nominal dataset $\mathcal{D}$ by adding and/or removing up to $n$ data-points.

To simplify our exposition below, we use $\mathcal{T}$ to denote cases where either the bounded or unbounded adversaries may be applied. We refer to $\mathcal{T}$ interchangeably as either the poisoning adversary, or set of potentially poisoned datasets.

**Poisoning Attack Goals.** Following the taxonomy of poisoning attacks from Tian et al. (2022), attacks can be broadly classified into three categories based on adversarial goals. *Untargeted poisoning* aims to degrade the overall performance of the model, potentially causing denial-of-service by corrupting training data, such as flipping sample labels or injecting noise into feature representations. In contrast, *targeted poisoning* attacks focus on manipulating the model's behavior for specific inputs while maintaining normal performance on others, effectively forcing misclassification of particular samples. A more advanced form of targeted poisoning is *backdoor attacks*, where adversaries implant hidden triggers in training data that activate only when specific patterns appear in test inputs, making the attack harder to detect. For each attack goal, one can formulate an adversarial objective function, which we aim to bound using our certification procedure. We discuss potential formulations and their certification in Appendix B.

## 3 Methodology

This section introduces a novel approach for certifying robustness against poisoning attacks by deriving sound bounds on the set of model parameters that can be reached under a given attack. We begin by demonstrating how robustness can be certified using such parameter-space bounds. Next, we present a general algorithm that bounds the effect of adversarial manipulations on model parameters, yielding an interval that encompasses all reachable training parameters. Finally, we instantiate our framework with a novel formulation of CROWN-style bounds, enabling the sound computation of the necessary quantities.

### 3.1 Parameter-Space Certificates of Robustness

The key concept behind our framework is to bound the parameters obtained via the training function $M(f, \theta', \mathcal{D})$ given the adversary $\mathcal{T}(\mathcal{D})$. Before detailing the method, we first formalize our definition of parameter-space bounds and how they can be translated into formal, provable guarantees on poisoning robustness.

**Definition 3.1** (Valid parameter-space bounds). An interval over parameters $[\theta^L, \theta^U]$ is a valid parameter-space bound on a poisoning adversary, $\mathcal{T}(\mathcal{D})$, if:

$$\theta = M(f, \theta', \tilde{\mathcal{D}}) \in [\theta^L, \theta^U] \qquad \forall \tilde{\mathcal{D}} \in \mathcal{T}(\mathcal{D}) \tag{1}$$

where the interval domain is interpreted element-wise.

Definition 3.1 captures a super-set of all the parameters the learning algorithm can reach given the intervention of an adversary defined by $\mathcal{T}(\mathcal{D})$. Next, we assume the existence of an objective function $J$ that the adversary attempts to maximize; we give concrete realizations of $J$ for well-studied adversarial goals including backdoor attacks in Appendix B. Regardless of the form of $J$ we have that:

**Theorem 3.2.** *Let $[\theta^L, \theta^U]$ be a valid parameter-space bound for a poisoning adversary $\mathcal{T}$. Then, for a given adversarial goal $J$, one can compute a sound upper bound (i.e. a certificate) by optimizing over the parameter space, rather than dataset space:*

$$\max_{\tilde{\mathcal{D}} \in \mathcal{T}} J\left(f^{M(f, \theta', \tilde{\mathcal{D}})}\right) \leq \max_{\tilde{\theta} \in [\theta^L, \theta^U]} J\left(f^{\tilde{\theta}}\right). \tag{2}$$

Theorem 3.2 enables us to translate our bounds on parameter-space bounds directly to bounds on the adversarial goals. Unlike the intractable maximization over datasets (left hand side of Equation (2)), the maximization over parameter space (right hand side of Equation (2)) can be efficiently upper-bounded using popular certification techniques (Adams et al., 2023; Wicker et al., 2020; 2023). The final result is a guarantee that *there does not exist a poisoning adversary who is more successful than the bound in Theorem 3.2.*

### 3.2 Abstract Gradient Training for Valid Parameter Space Bounds

In this section, we provide a high-level framework, which we term *Abstract Gradient Training* (AGT), for computing parameter bounds that respect Definition 3.1. Our framework is applicable to any first-order training algorithm[1], but here we choose to focus on vanilla SGD to ease our exposition. We consider the standard SGD update step:

$$\theta \leftarrow \theta - \alpha \frac{1}{|\mathcal{B}|} \sum_{(x,y) \in \mathcal{B}} \nabla_\theta \mathcal{L}\left(f^\theta(x), y\right) \tag{3}$$

where $\mathcal{B} \subseteq \mathcal{D}$ is the sampled batch at the current iteration. The function $M(f, \theta', \mathcal{D})$, in the simplest case, iteratively applies the update (3) for a fixed, finite number of iterations starting from $\theta = \theta'$. Therefore, to bound the effect of a poisoning attack, we iteratively apply bounds on update (3). In Algorithm 1, we present a general framework for computing parameter-space bounds for SGD given a poisoning adversary $\mathcal{T}$, and observe the following:

---

[1]Including those based on momentum or adaptive moments.

1. Lines 6-7 compute the standard SGD update rule.

2. Lines 8-10 compute bounds on the SGD update rule, taking into account poisoning of the current batch and *all previously seen batches*.

Starting from $\theta^L = \theta^U = \theta'$, the reachable parameter interval is maintained over every iteration of the algorithm, giving us the following result:

**Theorem 3.3.** *Algorithm 1 returns valid parameter-space bounds on a $T(\mathcal{D})$ poisoning adversary for stochastic gradient descent training procedure $M(f, \theta', \mathcal{D})$.*

Algorithm 1 allows us to bound the effect of a bounded adversary without modifying the original training procedure. On the other hand, to consider unbounded adversaries we must include the additional clipping operation, highlighted in purple. This limits the maximum contribution of any arbitrarily added or perturbed data-points, which is a requirement for computing closed bounds.

### 3.3 Bounding the Descent Direction.

The main challenge of Algorithm 1 is in bounding the set $\Delta\Theta$, which is the set of all possible descent directions at the given iteration under $\mathcal{T}$. In particular, $\widetilde{\mathcal{B}} \in \mathcal{T}(\mathcal{B})$ represents the effect of the adversary's perturbations on the *current* batch, while the reachable parameter interval $\tilde{\theta} \in \left[\theta^L, \theta^U\right]$ represents the worst-case effect of adversarial manipulations to all *previously seen* batches. Exactly computing the set $\Delta\Theta$ is not computationally tractable, so we instead seek over-approximate element-wise bounds that can be computed efficiently within the training loop.

Notationally, we will refer to *per-sample* gradient terms as $\delta^{(i)} = \nabla_\theta \mathcal{L}\left(f^\theta(x^{(i)}), y^{(i)}\right)$, and use the symbols $\delta_L^{(i)}$ and $\delta_U^{(i)}$ to represent sound bounds on these terms (we handle the computation of these terms in Section 3.4). We seek to soundly aggregate these per-sample gradient bounds to obtain an interval over-approximation $[\Delta\theta^L, \Delta\theta^U]$ of the descent direction $\Delta\theta$, which in turn is combined with the parameter interval $[\theta^L, \theta^U]$ at each training iteration. The exact aggregation mechanism depends on the form of the poisoning adversary. Below we present an efficient over-approximation of the descent direction for the bounded adversary in the following theorem[2].

**Theorem 3.4** (Bounding $\Delta\theta$ for a bounded adversary). *Let $\mathcal{B} = \left\{\left(x^{(i)}, y^{(i)}\right)\right\}_{i=1}^b$ be a batch of size $b$, and let the model parameters $\theta$ lie within the interval $[\theta^L, \theta^U]$. Given a bounded adversary specified by $\langle n, \epsilon, p, \nu, q \rangle$, the SGD parameter update $\Delta\theta = \frac{1}{b}\sum\limits_{\widetilde{\mathcal{B}}} \nabla_\theta \mathcal{L}\left(f^\theta(\tilde{x}^{(i)}), \tilde{y}^{(i)}\right)$ is bounded element-wise above and below by*

$$\Delta\theta^U = \frac{1}{b}\left(\underset{n}{\text{SEMax}}\left\{\tilde{\delta}_U^{(i)} - \delta_U^{(i)}\right\}_{i=1}^b + \sum_{i=1}^b \delta_U^{(i)}\right), \tag{4}$$

$$\Delta\theta^L = \frac{1}{b}\left(\underset{n}{\text{SEMin}}\left\{\tilde{\delta}_L^{(i)} - \delta_L^{(i)}\right\}_{i=1}^b + \sum_{i=1}^b \delta_L^{(i)}\right) \tag{5}$$

*for any batch $\widetilde{\mathcal{B}} \in T^{\langle n, \epsilon, p, \nu, q \rangle}(\mathcal{B})$. Here,*

- *$\delta_L^{(i)}$ and $\delta_U^{(i)}$ are sound bounds capturing the effect of previous adversarial manipulations, satisfying $\delta_L^{(i)} \leq \delta \leq \delta_U^{(i)}$ for all $\delta$ in the set*

$$\left\{\nabla_{\tilde{\theta}}\mathcal{L}\left(f^{\tilde{\theta}}(x^{(i)}), y^{(i)}\right) \mid \tilde{\theta} \in [\theta^L, \theta^U]\right\}. \tag{6}$$

- *$\tilde{\delta}_L^{(i)}$ and $\tilde{\delta}_U^{(i)}$ are bounds on the **combined** effect of previous poisoning and the worst-case adversarial perturbations of the i-th data-point in the current batch, satisfying $\tilde{\delta}_L^{(i)} \leq \tilde{\delta} \leq \tilde{\delta}_U^{(i)}$ for all $\tilde{\delta}$ in*

$$\left\{\nabla_{\tilde{\theta}}\mathcal{L}\left(f^{\tilde{\theta}}(\tilde{x}), \tilde{y}\right) \left|\begin{array}{c} \tilde{\theta} \in [\theta^L, \theta^U] \\ \|x^{(i)} - \tilde{x}\|_p \leq \epsilon \\ \|y^{(i)} - \tilde{y}\|_q \leq \nu \end{array}\right.\right\}. \tag{7}$$

---

[2]The analogous theorem for the unbounded adversary can be found in Appendix D.

The operations $\text{SEMax}_a$ and $\text{SEMin}_a$ correspond to taking the sum of the element-wise top/bottom-$a$ elements over each index of the input vectors[3]. This theorem, and its constituent operations, are discussed in more detail in Appendix F.4.

The update rule in Theorem 3.4 accounts for past poisoning via gradient bounds $(\delta_L^{(i)}, \delta_U^{(i)})$ valid for all reachable $\theta$. We then bound the impact of current-batch adversarial attacks by selecting the $n$ points exhibiting the worst-case gradient bounds under poisoning $(\tilde{\delta}_L^{(i)}, \tilde{\delta}_U^{(i)})$. To soundly bound the descent direction for all parameters, this update is applied independently to each parameter index. This approach, while computationally efficient for propagating interval enclosures, introduces a potentially loose over-approximation, as the $n$ worst-case points for one parameter index are unlikely to be the same for all indices.

### 3.4 Computation of Sound Gradient Bounds

This section presents a novel algorithm for bounding optimization problems of the form

$$\max \& \min_{\tilde{x}, \tilde{y}, \tilde{\theta}} \nabla_{\tilde{\theta}} \mathcal{L}\left(f^{\tilde{\theta}}\left(\tilde{x}\right), \tilde{y}\right) \quad \text{s.t.} \quad \begin{aligned} &\tilde{\theta} \in [\theta^L, \theta^U] \\ &\|x - \tilde{x}\|_p \leq \epsilon \\ &\|y - \tilde{y}\|_q \leq \nu \end{aligned} \quad . \tag{8}$$

Bounds on these optimization problems are exactly the per-sample gradient bounds $\tilde{\delta}_L, \tilde{\delta}_U$ required by Theorem 3.4. Noting that problems of the form (6) can be recovered by setting $\nu = \epsilon = 0$, we focus solely on this more general case in this section.

Computing the exact solution to (8) is, in general, a non-convex and NP-hard optimization problem (Katz et al., 2017). However, we require only over-approximate solutions; while these can introduce (potentially significant) looseness to the reachable parameter set, they will always maintain soundness. Future work could investigate exact solutions, e.g., via mixed-integer programming (Huchette et al., 2023; Tsay et al., 2021). Owing to their tractability, our discussion focuses on the novel linear bound propagation techniques we develop for abstract gradient training. We present only a brief overview of our approach in this section; full details can be found in Appendix C.

**Neural Networks.** While the algorithm presented in §3.2 is general to any machine learning model trained via stochastic gradient descent, we focus our discussion on neural network models for the remainder of the paper. We first define a neural network model $f^\theta : \mathbb{R}^{n_{\text{in}}} \to \mathbb{R}^{n_{\text{out}}}$ with $K$ layers and parameters $\theta = \left\{(W^{(i)}, b^{(i)})\right\}_{i=1}^K$ as:

$$\hat{z}^{(k)} = W^{(k)} z^{(k-1)} + b^{(k)}, \quad z^{(k)} = \sigma\left(\hat{z}^{(k)}\right)$$

where $z^{(0)} = x$, $f^\theta(x) = \hat{z}^{(K)}$, and $\sigma$ is the activation function, which we take to be ReLU.

**Certification of Neural Networks.** Solving problems of the form $\min \left\{\cdot \mid \|x - \tilde{x}\|_p \leq \epsilon\right\}$ for neural networks has been well-studied in the context of adversarial robustness certification. However, optimizing over inputs, labels and parameters, e.g., $\min \left\{\cdot \mid \tilde{\theta} \in [\theta^L, \theta^U], \|x - \tilde{x}\|_p \leq \epsilon, \|y - \tilde{y}\|_q \leq \nu\right\}$ is much less well-studied, and to-date similar problems have appeared primarily in the certification of probabilistic neural networks (Wicker et al., 2020). Our approach decomposes (8) into forward and backward passes through the neural network. We first compute bounds on the network's output for a given input, label, and parameter domain, then back-propagate these bounds to obtain gradient bounds.

**Interval Arithmetic.** For ease of exposition, we will represent interval matrices with bold symbols i.e., $\boldsymbol{A} := [A_L, A_U] \subset \mathbb{R}^{n_1 \times n_2}$. We define $\oplus, \otimes, \odot$ to represent interval matrix addition, matrix multiplication and element-wise multiplication, respectively, such that

$$\begin{aligned} A + B \in [\boldsymbol{A} \oplus \boldsymbol{B}] \quad &\forall A \in \boldsymbol{A}, B \in \boldsymbol{B}, \\ A \times B \in [\boldsymbol{A} \otimes \boldsymbol{B}] \quad &\forall A \in \boldsymbol{A}, B \in \boldsymbol{B}, \\ A \circ B \in [\boldsymbol{A} \odot \boldsymbol{B}] \quad &\forall A \in \boldsymbol{A}, B \in \boldsymbol{B}. \end{aligned}$$

---

[3]We assume that $n \leq b$. If $n > b$, we take SEMin/Max with respect to $\min(b, n)$ instead.

We denote interval vectors as $\boldsymbol{a} := [a_L, a_U]$ with analogous operations. We note that all matrix operations involving intervals can be computed using standard interval arithmetic techniques in at most $4\times$ the cost of a standard matrix operation, for example using Rump's algorithm Rump (1999). Interval arithmetic is commonly applied as a basic verification or adversarial training technique by propagating intervals through the intermediate layers of a neural network (Gowal et al., 2018).

**Forward Pass Bounds.** Mirroring developments in robustness certification of neural networks, we provide a novel, explicit extension of the CROWN algorithm (Zhang et al., 2018) to account for interval-bounded weights. The standard CROWN algorithm bounds the outputs of the $m$-th layer of a neural network by back-propagating linear bounds over each intermediate activation function to the input layer. We extend this framework to interval parameters, where the weights and biases involved in these linear relaxations are themselves intervals. We note that linear bound propagation with interval parameters has been studied previously in the context of floating-point sound certification (Singh et al., 2019). In the interest of space, we present only the upper bound of our extended CROWN algorithm here, with the full version presented in Appendix C.

**Proposition 3.5** (Explicit upper bounds of neural network $f$ with interval parameters). *Given an m-layer neural network function $f : \mathbb{R}^{n_{in}} \to \mathbb{R}^{n_{out}}$ whose unknown parameters lie in the intervals $b^{(k)} \in \boldsymbol{b}^{(k)}$ and $W^{(k)} \in \boldsymbol{W}^{(k)}$ for $k = 1, \ldots, m$, there exists an explicit function*

$$f_j^U\left(x, \Lambda^{(0:m)}, \Delta^{(1:m)}, b^{(1:m)}\right) = \Lambda_{j,:}^{(0)}x + \sum_{k=1}^{m}\Lambda_{j,:}^{(k)}\left(b^{(k)} + \Delta_{:,j}^{(k)}\right) \tag{9}$$

*such that $\forall x \in \boldsymbol{x}$*

$$f_j(x) \leq \max\left\{f_j^U\left(\cdot\right) \mid \Lambda^{(k)} \in \boldsymbol{\Lambda}^{(k)}, b^k \in \boldsymbol{b}^{(k)}\right\} \tag{10}$$

*where $\boldsymbol{x}$ is a closed input domain and $\Lambda^{(0:m)}, \Delta^{(1:m)}$ are the equivalent weights and biases of the linear upper bounds, respectively. The bias term $\Delta^{(1:m)}$ is explicitly computed based on the linear bounds on the activation functions. The weight $\Lambda^{(0:m)}$ lies in an interval $\boldsymbol{\Lambda}^{(0:m)}$ which is computed in an analogous way to standard, non-interval CROWN.*

Given the upper bound function $f_j^U(\cdot)$ defined above and intervals over all the relevant variables, we can compute the following closed-form global upper bound:

$$\gamma_j^U = \max\left\{\boldsymbol{\Lambda}_{j,:}^{(0)} \otimes \boldsymbol{x} \oplus \sum_{k=1}^{m}\boldsymbol{\Lambda}_{j,:}^{(k)} \otimes \left[\boldsymbol{b}^{(k)} \oplus \Delta_{:,j}^{(k)}\right]\right\}.$$

In the above, bold symbols are used to denote interval domains, with $\otimes$ and $\oplus$ being interval matrix multiplication and addition, respectively. Interval arithmetic, and its application to our CROWN-style bounds, is discussed in detail in Appendix C.

**Backward Pass Bounds.** Given bounds on the forward pass of the neural network, we can bound the backward pass (the gradients) of the model. We do this by extending the interval arithmetic based approach of Wicker et al. (2022) (which bounds derivatives of the form $\partial\mathcal{L}/\partial z^{(k)}$) to additionally bound the derivatives w.r.t. the parameters. In the below, we assume that sound bounds on the output logits $\hat{z}^{(K)}$ of the network have been obtained, e.g. using our CROWN-style bounds.

First, we back-propagate intervals over $y^\star$ (the label) and $\hat{z}^{(K)}$ (the logits) to compute an interval over $\boldsymbol{\partial\mathcal{L}/\partial\hat{z}^{(K)}}$, the gradient of the loss w.r.t. the logits of the network. The procedure for computing this interval is described in Appendix C for a selection of loss functions. We then use interval bound propagation to back-propagate this interval through the network to compute intervals over all gradients:

$$\frac{\partial\mathcal{L}}{\partial z^{(k-1)}} = \left(\boldsymbol{W^{(k)}}\right)^\top \otimes \frac{\partial\mathcal{L}}{\partial\hat{z}^{(k)}}, \quad \frac{\partial\mathcal{L}}{\partial\hat{z}^{(k)}} = \left[H\left(\hat{z}_L^{(k)}\right), H\left(\hat{z}_U^{(k)}\right)\right] \odot \frac{\partial\mathcal{L}}{\partial z^{(k)}}$$

$$\frac{\partial\mathcal{L}}{\partial\boldsymbol{W^{(k)}}} = \frac{\partial\mathcal{L}}{\partial\hat{z}^{(k)}} \otimes \left[\left(z_L^{(k-1)}\right)^\top, \left(z_U^{(k-1)}\right)^\top\right], \quad \frac{\partial\mathcal{L}}{\partial\boldsymbol{b^{(k)}}} = \frac{\partial\mathcal{L}}{\partial\hat{z}^{(k)}}$$

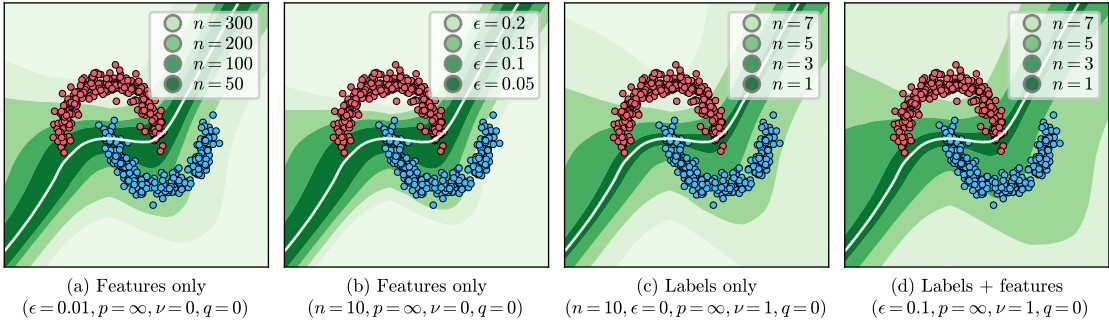

(a) Features only
$(\epsilon = 0.01, p = \infty, \nu = 0, q = 0)$

(b) Features only
$(n = 10, p = \infty, \nu = 0, q = 0)$

(c) Labels only
$(n = 10, \epsilon = 0, p = \infty, \nu = 1, q = 0)$

(d) Labels + features
$(\epsilon = 0.1, p = \infty, \nu = 1, q = 0)$

Figure 1: Regions where our certification holds for a classifier trained on the halfmoons dataset. The white line shows the decision boundary of the model, and each colored region indicates the area for which we *cannot* certify predictions for the given adversary.

where $H(\cdot)$ is the Heaviside function, and $\odot$ is the (interval) element-wise product. The resulting gradient intervals suffice to bound all solutions to our original optimization problem (8). That is, the gradients of the network lie within these intervals for all $W^{(k)} \in \boldsymbol{W^{(k)}}$, $b^{(k)} \in \boldsymbol{b^{(k)}}$, $\|x - x^\star\|_p \leq \epsilon$, and $\|y - y^\star\|_q \leq \nu$. These bounds are exactly those required to compute $\delta_L, \delta_U, \tilde{\delta}_L$, and $\tilde{\delta}_U$ needed to bound the descent direction in Theorem 3.4.

### 3.5 Algorithm Analysis and Discussion

Figure 1 visualizes the resulting worst-case decision boundaries for a simple binary classifier consisting of a neural network with a hidden layer of 128 neurons. In this classification setting, label poisoning results in looser bounds than feature space poisoning, with $n = 5$ producing bounds of approximately the same width as $n = 200$. This is due to the relatively large interval introduced by a label flipping attack $y^{(i)} \in \{0, 1\}$, compared to an interval of width $\epsilon$ introduced in a feature-space attack. We also emphasise that Algorithm 1 assumes at most $n$ poisoned points *per batch*, rather than per dataset. In regression settings, label poisoning is relatively weaker than feature poisoning for a given strength $\epsilon = \nu$, since the feature-space interval propagates through both the forward and backward training passes, while the label only participates in the backward pass. This effect is particularly pronounced in deep networks, since interval / CROWN bounds tend to weaken exponentially with depth (Mao et al., 2023; Sosnin & Tsay, 2024).

**Computing Certificates of Poisoning Robustness.** Algorithm 1 returns valid parameter-space bounds $[\theta_L, \theta_U]$ for a given poisoning adversary. To provide certificates of poisoning robustness for a specific query at a point $x$, we first bound the model output $f^\theta(x)$ for all $\theta \in [\theta_L, \theta_U]$ using the bound-propagation procedure described above. In classification settings, the robustness of the prediction can then be certified by checking if the lower bound on the output logit for the target class is greater than the upper bounds of all other classes (i.e. $[f_j^\theta(x)]_L \geq [f_i^\theta(x)]_U \forall i \neq j$). If this condition is satisfied, then the model always predicts class $j$ at the point $x$ for all parameters within our parameter-space bounds, and thus this prediction is certifiably robust to poisoning. Details on computing bounds on other adversary goals can be found in Appendix B.

**Comparison to Interval Bound Propagation.** The CROWN algorithm in §3.4 is not strictly tighter than interval bound propagation (IBP). Specifically, the non-associativity of double-interval matrix multiplication leads to significantly different interval sizes depending on the order in which the multiplications are performed: IBP performs interval matrix multiplications in a 'forwards' ordering, while CROWN uses a 'backwards' ordering. Empirically, we observe that CROWN tends to be tighter for deeper networks, while IBP may outperform CROWN for smaller networks. In our numerical experiments, we compute both CROWN and IBP bounds and take the element-wise tightest bound.

**Combined Forward and Backward Pass Bounds.** The CROWN algorithm can be applied to any composition of functions that can be upper- and lower-bounded by linear equations. Therefore, it is possible

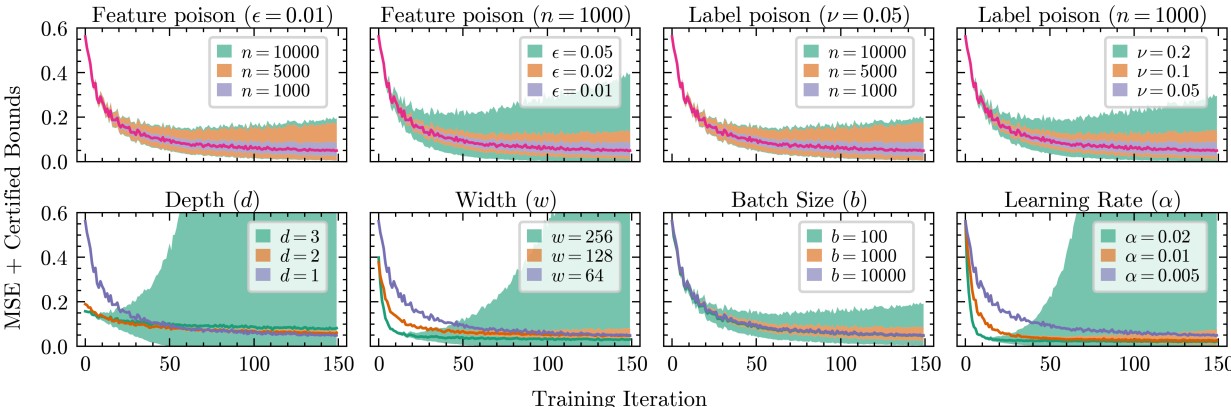

Figure 2: Mean squared error bounds on the UCI-houseelectric dataset for a bounded adversary. Top: Effect of adversary strength. Bottom: Effect of model/training hyperparameters (with $n = 100, \epsilon = 0.01, \nu = 0$). Where not stated, $p = q = \infty, d = 1, w = 50, b = 10000, \alpha = 0.02$.

to consider both the forwards and backwards passes in a single combined CROWN pass for many loss functions. However, linear bounds on the gradient of the loss function tend to be relatively loose, e.g., linear bounds on the softmax function may be orders-of-magnitude looser than constant $[0, 1]$ bounds (Wei et al., 2023). As a result, we found that the tightest bounds were obtained using IBP / CROWN on the forward pass and IBP on the backward pass.

**Computational Complexity.** The computational complexity of Algorithm 1 depends on the method used to bound on the gradients. In the simplest case, IBP can be used to compute bounds on the gradients in $4\times$ the cost of a standard forward and backward pass. Likewise, our CROWN bounds admit a cost of at most 4 times the cost of the original CROWN algorithm. For an $m$ layer network with $n$ neurons per layer and $n$ outputs, the time complexity of the original CROWN algorithm is $\mathcal{O}(m^2 n^3)$ (Zhang et al., 2018). We further note that the SEMin / SEMax operations can be computed in $\mathcal{O}(b)$ for each index and in practice can be efficiently parallelized using GPU-based implementations. In Appendix G, we observe that AGT typically incurs a runtime of less than $2\times$ standard training. In summary, Abstract Gradient Training using IBP has time complexity equivalent to standard neural network training ($\mathcal{O}(bmn^2)$ for each batch of size $b$), but with our tighter, CROWN-based, bounds the complexity is $\mathcal{O}(bm^2 n^3)$ per batch.

**Limitations.** While Algorithm 1 is able to obtain valid-parameter space bounds for any gradient-based training algorithm, the tightness of these bounds depends on the exact architecture, hyperparameters and training procedure used. In particular, bound-propagation between successive iterations of the algorithm assumes the worst-case poisoning at each parameter index *simultaneously*, which is not realizable by any practical poisoning attack. Therefore, obtaining non-vacuous guarantees with our algorithm often requires training with larger batch-sizes and/or for fewer epochs than is typical. Additionally, certain loss functions, such as multi-class cross entropy, have particularly loose interval relaxations. Therefore, AGT obtains relatively weaker guarantees for multi-class problems when compared to regression or binary classification settings. We hope that tighter bound-propagation approaches, such as those based on more expressive abstract domains, may overcome this limitation in future works.

## 4 Experiments

In this section we experimentally validate the effectiveness of our proposed approach. We provide complete details of hyper-parameters and run-times of our experiments in Appendix G. For classification tasks, we report the *certified accuracy*, which is our certified lower bound on the accuracy of any model poisoned with the given attack; likewise, certified mean squared error refers to an upper bound

on the loss in regression tasks. A code repository to reproduce our experiments can be found at: https://github.com/psosnin/AbstractGradientTraining.

**UCI Regression (Household Power Consumption).** We first consider a relatively simple regression model for the household electric power consumption ('houseelectric') dataset from the UCI repository (Hebrail & Berard, 2012) with fully connected neural networks and MSE as loss function. Figure 2 (top) shows the progression of the nominal and worst/best-case MSE (computed for the test set) for a $1 \times 50$ neural network and various parameterizations of poisoning attacks. As expected, we observe that increasing each of $n$, $\epsilon$, and $\nu$ results in looser performance bounds. We note that the setting of $n = 10000$ corresponds to 100% of the dataset being potentially poisoned.

Figure 2 (bottom) shows the progression of bounds on the MSE (computed for the test set) over the training procedure for a fixed poisoning attack ($n = 100, \epsilon = 0.01$) and various hyperparameters of the regression model. In general, we observe that increasing model size (width or depth) results in looser performance guarantees. As expected, increasing the batch size improves our bounds, as the number of potentially poisoned samples $n$ remains fixed and their worst-case effect is 'diluted'. Increasing the learning rate accelerates both the model training and the deterioration of the bounds.

**Comparison of CROWN vs IBP.** To evaluate the tightness of our proposed CROWN-based bounds relative to standard interval bound propagation, we use the UCI dataset setting described above to train models of varying sizes. We apply Algorithm 1 with either IBP or CROWN-based forward pass bounds and compare the resulting parameter space intervals. Figure 3 reports the average bound width at the end of training, computed as $\frac{1}{|\theta|} \sum_i (\theta_i^U - \theta_i^L)$. We find that our CROWN-based bounds yield tighter parameter intervals than IBP for deeper models. However, as previously discussed, CROWN is not strictly tighter in this setting and we observe that IBP can outperform CROWN for shallower but wider architectures.

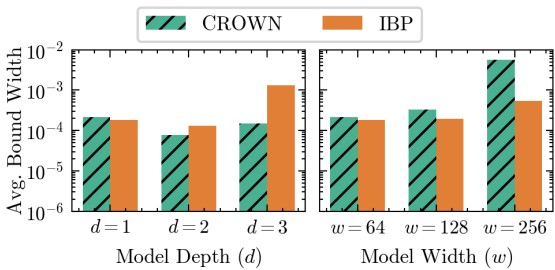

Figure 3: Comparison of AGT with CROWN vs IBP bounds for various model sizes trained on the UCI dataset ($d = 1$ and $w = 64$ where not stated).

Accordingly, in all other experiments, we compute both sets of bounds at each training iteration and take their element-wise minimum to obtain the tightest parameter intervals.

**Baseline Comparison.** We compare our approach with DPA Levine & Feizi (2020) and randomized smoothing (RS) Rosenfeld et al. (2020) on MNIST. To make the approaches comparable, we only consider a label-flipping attack ($q = 0, \nu = 1$) on the MNIST dataset. In label-only poisoning settings, it is common to use unsupervised learning approaches on the (assumed clean) features prior to training a classification model (e.g. SS-DPA (Levine & Feizi, 2020)). Therefore, we first project the data into a 32-dimensional feature space using PCA. Figure 4 illustrates the certified accuracy using this methodology. Before discussing the quantitative comparison, we highlight the significant qualitative differences between the baselines and our approach. Randomized smoothing only applies to linear models and the label flipping regime whereas our approach is general. DPA applies to generalized poisoning adversaries, but requires both substantial overhead (relative to AGT) and modification of the learning algorithm. In Figure 4, DPA uses an ensemble of 3000

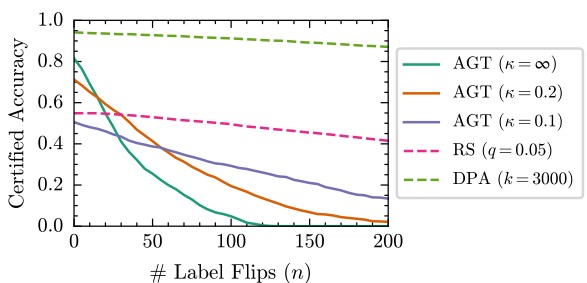

Figure 4: Certified accuracy on the MNIST dataset under a label-flipping attack. Curves for Deep Partition Aggregation (DPA) and Randomized Smoothing (RS) reproduced from Figures 1 of Levine & Feizi (2020) and Rosenfeld et al. (2020), respectively.

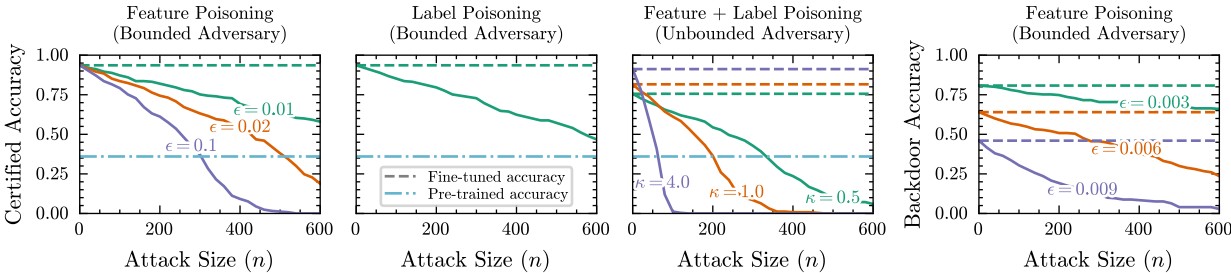

Figure 5: Certified accuracy (left) and backdoor accuracy (right) for a binary classifier fine-tuned on the Drusen class of OCTMNIST for an attack size up to 10% poisoned data per batch ($b = 6000, p = \infty, q = 0, \nu = 1$). Dashed lines show the nominal accuracy of each fine-tuned model.

models thus incurring 3000 times the space costs compared to normal training. Quantitatively, AGT can outperform randomized smoothing, but only for a small number of label flips. However, there is no setting in which AGT outperforms DPA.

**MedMNIST Image Classification.** Next, we consider fine-tuning a classifier trained on the retinal OCT dataset (OCTMNIST) (Yang et al., 2021), which contains four classes—one normal and three abnormal. The dataset is unbalanced, and we consider the simpler normal vs abnormal binary classification setting. We consider the 'small' architecture from Gowal et al. (2018), comprising two convolutional layers of width 16 and 32 and a dense layer of 100 nodes, and the following fine-tuning scenario: the model is first pre-trained without the rarest class (Drusen) using the robust training procedure from Wicker et al. (2022), so that the resulting model is robust to feature perturbations during fine-tuning. We then assume Drusen samples may be poisoned and add them as a new abnormal class to fine-tune the dense layer, with a mix of 50% Drusen samples ($b = 6000$ with 3000 Drusen) per batch.

Figure 5 shows how increasing the strength of the potential poisoning attack worsens the bound on prediction accuracy. With feature-only poisoning, a poisoning attack greater than $\epsilon = 0.02$ over $n \approx 500$ samples produces bounds worse than the prediction accuracy of the original pre-trained model. With an unbounded label and feature poisoning adversary, the bounds are weaker, as expected. Higher certified accuracy can be obtained by increasing the clipping parameter $\kappa$, at the cost of nominal model accuracy (Figure 5, center right). With label-only poisoning, the certificates are relatively stronger, as the training procedure requires approximately $n \geq 600$ poisoned samples for the prediction accuracy bound to reach the original pre-trained model's accuracy. The setting of $n = 600$ corresponds to 20% of the Drusen data per batch being mis-labeled.

Finally, we consider a backdoor attack setting where the $\epsilon$ used at training and inference times is the same. The model is highly susceptible to adversarial perturbations at inference time even without data poisoning, requiring only $\epsilon = 0.009$ to reduce the certified backdoor accuracy to $< 50\%$. As the strength of the adversary increases, the accuracy that we are able to certify decreases. We note that tighter verification algorithms can be applied at inference time to obtain stronger guarantees.

**Fine-Tuning PilotNet.** Finally, we fine-tune a model that predicts steering angles for autonomous driving given an input image (Bojarski et al., 2016). The model contains convolutional layers of 24, 36, 48, and 64 filters, followed by fully connected layers of 100, 50, and 10 nodes. The fine-tuning setting is similar to above: first, we pre-train the model on videos 2–6 of the Udacity self-driving car dataset[4]. We then fine-tune the dense layers on video 1 (challenging lighting conditions) assuming potential label poisoning.

Figure 6 shows the bounds on mean squared error for the video 1 data and visualizes how the bounds translate to the predicted steering angle. We again see that fine-tuning improves accuracy on the new data,

---

[4]`github.com/udacity/self-driving-car/tree/master`

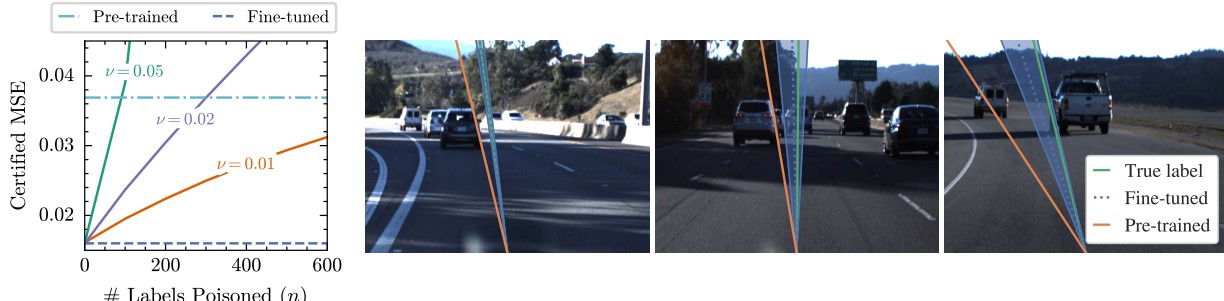

Figure 6: Left: Fine-tuning PilotNet on unseen data with a bounded label poisoning attack ($q = \infty$). Right: We plot the steering angle prediction before and after fine-tuning ($n = 300, q = \infty, \nu = 0.01$). The angle of the lines indicates the predicted steering angle.

but also that the MSE bounds deteriorate as the number of potentially poisoned samples increases (Figure 6, left). The rate of deterioration depends strongly on poisoning strength $\nu$.

## 5  Conclusions

We proposed a mathematical framework for computing sound parameter-space bounds on the influence of a poisoning attack for gradient-based training. Our framework defines generic constraint sets to represent general poisoning attacks and propagates them through the forward and backward passes of model training. Based on the resulting parameter-space bounds, we provided rigorous bounds on the effects of various poisoning attacks. Finally, we demonstrated our proposed approach to be effective on tasks including autonomous driving and medical image classification.

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

# A Related Works

**Data Poisoning.** Poisoning attacks have existed for nearly two decades and are a serious security concern (Biggio & Roli, 2018; Biggio et al., 2014; Newsome et al., 2006). In Muñoz-González et al. (2017) the authors formulate a general gradient-based attack that generates poisoned samples that corrupt model performance when introduced into the dataset (now termed, untargeted attack). Backdoor attacks manipulate a small proportion of the data such that, when a specific pattern is seen at test-time, the model returns a specific, erroneous prediction Chen et al. (2017); Gu et al. (2017); Han et al. (2022); Zhu et al. (2019). Popular defenses are attack specific, e.g., generating datasets using known attack strategies to classify and reject potentially poisoned inputs (Li et al., 2020). Alternative strategies apply noise or clipping to mitigate certain attacks (Hong et al., 2020).

**Poisoning Defenses.** General defenses to poisoning attacks seek to provide upper-bounds on the effectiveness of *any* attack strategy. In this area, Steinhardt et al. (2017) provide such upper-bounds for linear models trained with gradient descent. Rosenfeld et al. (2020) present a certified upper-bound on the effectiveness of $\ell_2$ perturbations on training labels for linear models using randomized smoothing. Xie et al. (2022) observe that differential privacy, which usually covers addition or removal of data points, can also provide statistical guarantees in some limited poisoning settings.

**Certified Poisoning Robustness** Relative to inference-time adversarial robustness (also referred to as evasion attacks), less attention has been devoted to provable guarantees against data poisoning adversaries. Existing methods for deterministic certification of robustness to poisoning adversaries involve design of a learning process with careful partitioning and ensembling such that the resulting model has poisoning robustness guarantees (Levine & Feizi, 2020; Wang et al., 2022; Rezaei et al., 2023). We refer to these methods as "aggregation" methods. In contrast, our approach is a method for analysis and certification of standard, unmodified machine learning algorithms. Aggregation approaches have been shown to offer strong guarantees against poisoning adversaries albeit at a substantial computational cost including: storing and training thousands of models on (potentially disjoint) subsets of the dataset and the requirement to evaluate each of the potentially thousands of models for each prediction; additionally, these methods require that one have potentially thousands of times more data than is necessary for training a single classifier. By designing algorithms to be robust to poisoning adversaries aggregation based approaches are able to scale to larger models than are considered in this work (Levine & Feizi, 2020; Wang et al., 2022; Rezaei et al., 2023). Yet, the computational cost of our approach, in the simplest case, is only four times that of standard training and inference and we do not require that one has access to enough data to train multiple well-performing models. Furthermore, our approach enables reasoning about backdoor attacks, where these partitioning approaches cannot. We finally highlight that the method presented in this paper is orthogonal/complementary to the partitioning approach, and thus future works may be able to combine the two effectively.

A concurrent work (Gosch et al., 2024) also develops certificates for neural networks under data poisoning by leveraging the Neural Tangent Kernel to establish an equivalence of a neural network with a Support Vector Machine. In contrast to this work, the certificates presented in (Gosch et al., 2024) are deterministic only under the assumption of infinite-width networks, and can only handle a restricted set of threat models compared to those described in Section 2.

**Certified Adversarial Robustness.** Sound algorithms (i.e., no false positives) for upper-bounding the effectiveness of inference-time adversaries are well-studied for trained models (Gehr et al., 2018) and training models for robustness (Gowal et al., 2018; Müller et al., 2022). These approaches typically utilize ideas from formal methods (Katz et al., 2017; Wicker et al., 2018) or optimization (Botoeva et al., 2020; Bunel et al., 2018; Huchette et al., 2023). Most related to this work are strategies that consider intervals over both model inputs and parameters (Wicker et al., 2020), as well as some preliminary work on robust explanations that bound the input gradients of a model Wicker et al. (2022). Despite these methodological relationships, none of these methods directly apply to the general training setting studied here.

# B  Poisoning Attack Goals and Certified Bounds

In this appendix, we define common poisoning attack objectives and demonstrate how our certification procedure can be used to bound the success of such attacks. We focus on the three primary objectives introduced in Section 2, and note that this is not an exhaustive list of adversarial goals.

## B.1  Poisoning Attack Goals

**Untargeted Poisoning.** Untargeted attacks aim to prevent training convergence, leading to an unusable model and denial of service (Tian et al., 2022). We formulate the goal with respect to some dataset of interest $\{(x^{(i)}, y^{(i)})\}_{i=1}^k$. In the case of denial of service, this is usually taken to be the training dataset, or some subset therein. The denial of service attack objective can thus be written as

$$\max_{\mathcal{D}' \in \mathcal{T}} \frac{1}{k} \sum_{i=1}^{k} \mathcal{L}\big(f^{M(f, \theta', \mathcal{D}')}(x^{(i)}), y^{(i)}\big) \tag{11}$$

where $\mathcal{L}$ is taken to be the training loss function. We can certify robustness to this kind of attack using a sound upper bound on this optimization problem.

**Targeted Poisoning.** Targeted poisoning is more task-specific and is typically evaluated over the test dataset. Rather than simply attempting to increase the loss, the adversary seeks to make model predictions fall outside a 'safe' set of outputs, $S(x^{(i)}, y^{(i)})$. In the simplest case, this can be the defined as the set of predictions matching the ground truth. The safe set can be more specific however, i.e., mistaking a lane marking for a person is safe, but not vice versa. The adversary's objective is given by:

$$\max_{\mathcal{D}' \in \mathcal{T}} \frac{1}{k} \sum_{i=1}^{k} \mathbb{1}\big(f^{M(f, \theta', \mathcal{D}')}(x^{(i)}) \notin S(x^{(i)}, y^{(i)})\big) \tag{12}$$

As before, a sound upper bound on (12) bounds the success rate of any targeted poisoning attacker. Note that with $k = 1$ we recover the pointwise certificate setting studied by Rosenfeld et al. (2020). This setting also covers 'unlearnable examples', such as the attacks considered by Huang et al. (2021).

**Backdoor Poisoning.** Backdoor attacks deviate from the above attacks by assuming that test-time data can be altered, via a so-called trigger manipulation. By assuming that the trigger manipulation(s) are bounded to a set $V(x)$ (e.g., an $\ell_\infty$ ball around the input), one can formulate the backdoor attack's goal as producing predictions outside a safe set $S(x^{(i)}, y^{(i)})$ (defined as before) for manipulated inputs:

$$\max_{\mathcal{D}' \in \mathcal{T}} \frac{1}{k} \sum_{i=1}^{k} \mathbb{1}\big(\exists x^\star \in V(x^{(i)}) \ s.t. \ f^{M(f, \theta', \mathcal{D}')}(x^\star) \notin S(x^{(i)}, y^{(i)})\big) \tag{13}$$

Backdoor attacks typically aim to leave the model performance on 'clean' data unchanged, which can be modeled via the data-dependent safe set $S$. Any sound upper bound to the above is a sound bound on the success rate of any backdoor attacker.

## B.2  Bounding Poisoning Attack Goals

In this section, we describe a procedure for computing bounds on each of the poisoning adversary's objectives. We recall from Theorem 3.2 that optimization problems with respect to an adversary $\mathcal{T}$ can be equivalently bounded by optimizing over a valid parameter space bound $[\theta^L, \theta^U]$ returned from Algorithm 1.

Given a test set $\{(x^{(i)}, y^{(i)})\}_{i=1}^{k}$, we will write the poisoning objectives above in the following form,

$$J\left(f^{\theta}\right) = \max_{\mathcal{D}' \in \mathcal{T}} \frac{1}{k} \sum_{i=1}^{k} g\left(f^{M(f,\theta',\mathcal{D}')}, x^{(i)}, y^{(i)}\right) \tag{14}$$

$$\leq \max_{\theta \in [\theta^L, \theta^U]} \frac{1}{k} \sum_{i=1}^{k} g\left(f^{\theta}, x^{(i)}, y^{(i)}\right) \tag{15}$$

$$\leq \frac{1}{k} \sum_{i=1}^{k} \max_{\theta \in [\theta^L, \theta^U]} g\left(f^{\theta}, x^{(i)}, y^{(i)}\right) \tag{16}$$

where the function $g(\cdot)$ is the adversary objective for a single sample. The first inequality holds by Theorem 3.2, and the second holds by relaxing the maximization over the summation. Thus, to bound the original poisoning objectives (11), (12), and (13), it suffices to compute bounds on the required quantity for each test sample independently.

**Certified Predictions and Backdoor Robustness.** Computing an bounds on

$$\max_{\theta^{\star} \in [\theta^L, \theta^U]} \mathbb{1}\left(f^{\theta^{\star}}(x^{(i)}) \notin S\right) \tag{17}$$

corresponds to checking if $f^{\theta^{\star}}(x^{(i)})$ lies within the safe set $S$ for all $\theta^{\star} \in [\theta^L, \theta^U]$. As before, we first compute bounds $f^L, f^U$ on $f^{\theta^{\star}}(x^{(i)})$ using our CROWN-based bounds. Given these bounds and assuming a multiclass classification setting, the predictions *not* reachable by any model within $[\theta^L, \theta^U]$ are those whose logit upper bounds lie below the logit lower bound of any other class. That is, the set of possible predictions $S'$ is given by

$$S' = \left\{i \text{ s.t. } \nexists j : f_i^U \leq f_j^L\right\}. \tag{18}$$

If $S' \subseteq S$, then $\max_{\theta^{\star}} \mathbb{1}\left(f^{\theta^{\star}}(x^{(i)}) \notin S\right) = 0$.

Backdoor attack robustness is computed in an analogous way, with the only difference being the logit bounds $f^L, f^U$ being computed over all $x \in V(x)$ and $\theta^{\star} \in [\theta^L, \theta^U]$. This case is also computed via our CROWN-based bound propagation.

**Denial of Service.** In the case of denial of service, it suffices to bound the training loss, such as the cross entropy and mean-squared-error losses. For cross entropy loss, one can compute the loss over the worst-case logits given bounds on the neural network output (from our CROWN algorithm). For mean squared error, a valid upper bound is obtained by computing the loss with respect the point in the output bounds that is furthest from the true label. Bounding these loss functions is discussed in more detail in Appendix C.

## C    Bound Propagation in Abstract Gradient Training

In this appendix, we provide a comprehensive overview of the bound-propagation procedure used to obtain gradient bounds in Abstract Gradient Training. We recall our aim of bounding (8), which we state explicitly for neural networks below:

$$
\begin{aligned}
\max_{\tilde{x}, \tilde{y}, W^{(1:K)}, b^{(1:K)}} \text{ or } \min \quad & \nabla_\theta \mathcal{L}\left(f\left(\tilde{x}\right), \tilde{y}\right) \\
\text{s.t.} \quad & W_L^{(k)} \leq W^{(k)} \leq W_U^{(k)}, & k = 1, \ldots, K \\
& b_L^{(k)} \leq b^{(k)} \leq b_U^{(k)}, & k = 1, \ldots, K \\
& \hat{z}^{(k)} = W^{(k)} z^{(k-1)} + b^{(k)}, & k = 1, \ldots, K \\
& z^{(k)} = \sigma\left(\hat{z}^{(k)}\right), & k = 1, \ldots, K \\
& \|x - \tilde{x}\|_p \leq \epsilon, \\
& \|y - \tilde{y}\|_q \leq \nu, \\
& f(x) = \hat{z}^{(K)}, \\
& z^{(0)} = x.
\end{aligned}
\tag{19}
$$

where the min or maximization problem is considered element-wise. To efficiently bound this problem, we start by decomposing it into distinct sub-problems, which we bound independently. In particular, we first aim to bound the forward pass variables $z^{(k)}$ using our CROWN-style bounds, and then back-propagate these bounds using interval propagation. Each sub-problem considers only a sub-set of constraints, so the resulting bounds are a valid over-approximation of the solutions to (19).

In the remainder of this section, we first provide an introduction to interval matrix arithmetic, which forms the foundation of our instantiation of AGT. We then present our novel CROWN-based bounds for bounding the forward pass through the neural network. Finally, we describe the interval bound propagation method used to back-propagate these bounds and compute the required gradient bounds.

### C.1 Interval Matrix Arithmetic

Here we provide a basic introduction to interval matrix arithmetic, which forms the basic building block of our CROWN-style bounds. For ease of exposition, we will represent interval matrices with bold symbols i.e., $\boldsymbol{A} := [A_L, A_U] \subset \mathbb{R}^{n_1 \times n_2}$. We denote interval vectors as $\boldsymbol{a} := [a_L, a_U]$ with analogous operations.

**Definition C.1** (Interval Matrix Arithmetic). Let $\boldsymbol{A} = [A_L, A_U]$ and $\boldsymbol{B} = [B_L, B_U]$ be intervals over matrices. Let $\oplus, \otimes, \odot$ represent interval matrix addition, matrix multiplication and elementwise multiplication, such that

$$
\begin{aligned}
A + B &\in [\boldsymbol{A} \oplus \boldsymbol{B}] & \forall A \in \boldsymbol{A}, B \in \boldsymbol{B}, \\
A \times B &\in [\boldsymbol{A} \otimes \boldsymbol{B}] & \forall A \in \boldsymbol{A}, B \in \boldsymbol{B}, \\
A \circ B &\in [\boldsymbol{A} \odot \boldsymbol{B}] & \forall A \in \boldsymbol{A}, B \in \boldsymbol{B}.
\end{aligned}
$$

These operations can be computed using standard interval arithmetic techniques in at most $4\times$ the cost of a standard matrix operation, for example using Rump's algorithm Rump (1999). Interval arithmetic is commonly applied as a basic verification or adversarial training technique by propagating intervals through the intermediate layers of a neural network (Gowal et al., 2018).

### C.2 Forward pass bounds: CROWN with Interval Parameters

In this section, we present our full extension of the CROWN algorithm (Zhang et al., 2018) for neural networks with interval parameters. The standard CROWN algorithm bounds the outputs of the $m$-th layer of a neural network by back-propagating linear bounds over each intermediate activation function to the input layer. We extend this framework to interval parameters, where the weights and biases involved in these linear relaxations are themselves intervals. We note that linear bound propagation with interval parameters has been studied previously in the context of floating-point sound certification (Singh et al., 2019).

Here, we present an explicit instantiation of the CROWN algorithm for interval parameters, which we recall from Section 3.4 in its complete form.

**Proposition 3.5** (Explicit output bounds of neural network $f$ with interval parameters). *Given an $m$-layer neural network function $f : \mathbb{R}^{n_{in}} \to \mathbb{R}^{n_{out}}$ whose unknown parameters lie in the intervals $b^{(k)} \in \boldsymbol{b}^{(k)}$ and $W^{(k)} \in \boldsymbol{W}^{(k)}$ for $k = 1, \ldots, m$, there exist two explicit functions*

$$
f_j^L \left( x, \Omega^{(0:m)}, \Theta^{(1:m)}, b^{(1:m)} \right) = \Omega_{j,:}^{(0)} x + \sum_{k=1}^m \Omega_{j,:}^{(k)} \left( b^{(k)} + \Theta_{:,j}^{(k)} \right) \tag{20}
$$

$$
f_j^U \left( x, \Lambda^{(0:m)}, \Delta^{(1:m)}, b^{(1:m)} \right) = \Lambda_{j,:}^{(0)} x + \sum_{k=1}^m \Lambda_{j,:}^{(k)} \left( b^{(k)} + \Delta_{:,j}^{(k)} \right) \tag{21}
$$

*such that $\forall x \in \boldsymbol{x}$*

$$
\begin{aligned}
f_j(x) &\geq \min \left\{ f_j^L \left( x, \Omega^{(0:m)}, \Theta^{(1:m)}, b^{(1:m)} \right) \mid \Omega^{(k)} \in \boldsymbol{\Omega}^{(k)}, b^k \in \boldsymbol{b}^{(k)} \right\} \\
f_j(x) &\leq \max \left\{ f_j^U \left( x, \Lambda^{(0:m)}, \Delta^{(1:m)}, b^{(1:m)} \right) \mid \Lambda^{(k)} \in \boldsymbol{\Lambda}^{(k)}, b^k \in \boldsymbol{b}^{(k)} \right\}
\end{aligned}
$$

*where $\boldsymbol{x}$ is a closed input domain and $\Lambda^{(0:m)}, \Delta^{(1:m)}, \Omega^{(0:m)}, \Theta^{(1:m)}$ are the equivalent weights and biases of the upper and lower linear bounds, respectively. The bias terms $\Delta^{(1:m)}, \Theta^{(1:m)}$ are explicitly computed based on the linear bounds on the activation functions. The weights $\Lambda^{(0:m)}, \Omega^{(0:m)}$ lie in intervals $\boldsymbol{\Lambda}^{(0:m)}, \boldsymbol{\Omega}^{(0:m)}$ which are computed in an analogous way to standard (non-interval) CROWN.*

**Computing Equivalent Weights and Biases.** Our instantiation of the CROWN algorithm in Proposition 3.5 relies on the computation of the equivalent bias terms $\Delta^{(1:m)}, \Theta^{(1:m)}$ and interval enclosures over the equivalent weights $\Omega^{(0:m)}, \Lambda^{(0:m)}$. This proceeds similarly to the standard CROWN algorithm but now accounting for intervals over the parameters $b^{(1:m)}, W^{(1:m)}$ of the network.

The standard CROWN algorithm bounds the outputs of the $m$-th layer of a neural network by back-propagating linear bounds over each intermediate activation function to the input layer. In the case of interval parameters, the sign of a particular weight may be ambiguous (when the interval spans zero), making it impossible to determine which linear bound to back-propagate. In such cases, we propagate a concrete bound for that neuron instead of its linear bounds.

When bounding the $m$-th layer of a neural network, we assume that we have pre-activation bounds $\hat{z}^{(k)} \in \left[ l^{(k)}, u^{(k)} \right]$ on all previous layers on the network. Given such bounds, it is possible to form linear bounds on any non-linear activation function in the network. For the $r$-th neuron in $k$-th layer with activation function $\sigma(z)$, we define two linear functions

$$h_{L,r}^{(k)}(z) = \alpha_{L,r}^{(k)} \left( z + \beta_{L,r}^{(k)} \right), \quad h_{U,r}^{(k)}(z) = \alpha_{U,r}^{(k)} \left( z + \beta_{U,r}^{(k)} \right)$$

such that $h_{L,r}^{(k)}(z) \leq \sigma(z) \leq h_{U,r}^{(k)}(z) \ \forall z \in \left[ l_r^{(k)}, u_r^{(k)} \right]$. The coefficients $\alpha_{U,r}^{(k)}, \alpha_{L,r}^{(k)}, \beta_{U,r}^{(k)}, \beta_{L,r}^{(k)} \in \mathbb{R}$ are readily computed for many common activation functions (Zhang et al., 2018).

Given the pre-activation and activation function bounds, the interval enclosures over the weights $\Omega^{(0:m)}, \Lambda^{(0:m)}$ are computed via a back-propagation procedure. The back-propagation is initialised with $\boldsymbol{\Omega}^{(m)} = \boldsymbol{\Lambda}^{(m)} = [I^{n_m}, I^{n_m}]$ and proceeds as follows:

$$\boldsymbol{\Lambda}^{(k-1)} = \left( \boldsymbol{\Lambda}^{(k)} \otimes \mathbf{W}^{(k)} \right) \odot \lambda^{(k-1)}, \quad \lambda_{j,i}^{(k)} = \begin{cases} \alpha_{U,i}^{(k)} & \text{if } k \neq 0, \ 0 \leq \left[ \boldsymbol{\Lambda}^{(k+1)} \otimes \mathbf{W}^{(k+1)} \right]_{j,i} \\ \alpha_{L,i}^{(k)} & \text{if } k \neq 0, \ 0 \geq \left[ \boldsymbol{\Lambda}^{(k+1)} \otimes \mathbf{W}^{(k+1)} \right]_{j,i} \\ 0 & \text{if } k \neq 0, \ 0 \in \left[ \boldsymbol{\Lambda}^{(k+1)} \otimes \mathbf{W}^{(k+1)} \right]_{j,i} \\ 1 & \text{if } k = 0. \end{cases}$$

$$\boldsymbol{\Omega}^{(k-1)} = \left( \boldsymbol{\Omega}^{(k)} \otimes \mathbf{W}^{(k)} \right) \odot \omega^{(k-1)}, \quad \omega_{j,i}^{(k)} = \begin{cases} \alpha_{L,i}^{(k)} & \text{if } k \neq 0, \ 0 \leq \left[ \boldsymbol{\Omega}^{(k+1)} \otimes \mathbf{W}^{(k+1)} \right]_{j,i} \\ \alpha_{U,i}^{(k)} & \text{if } k \neq 0, \ 0 \geq \left[ \boldsymbol{\Omega}^{(k+1)} \otimes \mathbf{W}^{(k+1)} \right]_{j,i} \\ 0 & \text{if } k \neq 0, \ 0 \in \left[ \boldsymbol{\Omega}^{(k+1)} \otimes \mathbf{W}^{(k+1)} \right]_{j,i} \\ 1 & \text{if } k = 0. \end{cases}$$

where we use $0 \leq [\cdot]$ and $0 \geq [\cdot]$ to denote that an interval is strictly positive or negative, respectively.

Finally, the bias terms $\Delta^{(k)}, \Theta^{(k)}$ for all $k < m$ can be computed as

$$\Delta_{i,j}^{(k)} = \begin{cases} \beta_{U,i}^{(k)} & \text{if } 0 \leq \left[ \boldsymbol{\Lambda}^{(k+1)} \otimes \mathbf{W}^{(k+1)} \right]_{j,i} \\ \beta_{L,i}^{(k)} & \text{if } 0 \geq \left[ \boldsymbol{\Lambda}^{(k+1)} \otimes \mathbf{W}^{(k+1)} \right]_{j,i} \\ u^{(k)} & \text{if } 0 \in \left[ \boldsymbol{\Lambda}^{(k+1)} \otimes \mathbf{W}^{(k+1)} \right]_{j,i} \end{cases}, \Theta_{i,j}^{(k)} = \begin{cases} \beta_{L,i}^{(k)} & \text{if } 0 \leq \left[ \boldsymbol{\Omega}^{(k+1)} \otimes \mathbf{W}^{(k+1)} \right]_{j,i} \\ \beta_{U,i}^{(k)} & \text{if } 0 \geq \left[ \boldsymbol{\Omega}^{(k+1)} \otimes \mathbf{W}^{(k+1)} \right]_{j,i} \\ l^{(k)} & \text{if } 0 \in \left[ \boldsymbol{\Omega}^{(k+1)} \otimes \mathbf{W}^{(k+1)} \right]_{j,i} \end{cases}$$

with the $m$-th bias terms given by $\Theta_{i,j}^{(m)} = \Delta_{i,j}^{(m)} = 0$.

**Closed-Form Global Bounds.** Given the two functions $f_j^L(\cdot)$, $f_j^U(\cdot)$ as defined above and intervals over all the relevant variables, we can compute the following closed-form global bounds:

$$\gamma_j^L = \min \left\{ \mathbf{\Omega}_{j,:}^{(0)} \otimes \boldsymbol{x} \oplus \sum_{k=1}^{m} \mathbf{\Omega}_{j,:}^{(k)} \otimes \left[ \boldsymbol{b}^{(k)} \oplus \Theta_{:,j}^{(k)} \right] \right\}$$

$$\gamma_j^U = \max \left\{ \mathbf{\Lambda}_{j,:}^{(0)} \otimes \boldsymbol{x} \oplus \sum_{k=1}^{m} \mathbf{\Lambda}_{j,:}^{(k)} \otimes \left[ \boldsymbol{b}^{(k)} \oplus \Delta_{:,j}^{(k)} \right] \right\}$$

where min / max are performed element-wise and return the lower / upper bounds of each interval enclosure. Then, we have $\gamma_j^L \leq f_j(x) \leq \gamma_j^U$ for all $x \in \boldsymbol{x}$, $b^{(k)} \in \boldsymbol{b}^{(k)}$ and $W^{(k)} \in \boldsymbol{W}^{(k)}$, which suffices to bound the output of the neural network as required to further bound the gradient of the network.

### C.3 Bounding the Backward Pass: Interval Bound Propagation

Turning to the backward pass, we first show how to bound the first partial derivative of the loss function given bounds from the forward pass. We then provide an interval back-propagation procedure for computing intervals over all gradients of the model.

**Loss Function Bounds.** Here we present the computation of bounds on the first partial derivative of the loss function required for Algorithm 1. In particular, we consider bounding the following optimization problem via interval arithmetic:

$$\min \& \max \left\{ \partial \mathcal{L}\left(y^{\star}, y'\right) / \partial y^{\star} \mid y^{\star} \in \left[y^L, y^U\right], \|y' - y^t\|_q \leq \nu \right\}$$

for some loss function $\mathcal{L}$ where $[y^L, y^U]$ are bounds on the logits of the model (obtained via the bound-propagation procedure), $y^t$ is the true label, and $y'$ is the poisoned label.

**Mean Squared Error Loss.** Taking $\mathcal{L}\left(y^{\star}, y'\right) = \|y^{\star} - y'\|_2^2$ to be the squared error and considering the $q = \infty$ norm, the required bounds are given by:

$$\partial l^L = 2\left(y^L - y^t - \nu\right)$$
$$\partial l^U = 2\left(y^U - y^t + \nu\right)$$

The loss itself can be upper-bounded by $l^U = \max\{(y^L - y^t)^2, (y^U - y^t)^2\}$ and lower bounded by

$$l^L = \begin{cases} 0 & \text{if } y^t \in [y^L, y^U] \\ \min\{(y^L - y^t)^2, (y^U - y^t)^2\} & \text{otherwise} \end{cases} \tag{22}$$

**Cross Entropy Loss.** To bound the gradient of the cross entropy loss, we first bound the output probabilities $p_i = [\sum_j \exp\left(y_j^{\star} - y_i^{\star}\right)]^{-1}$ obtained by passing the logits through the softmax function:

$$p_i^L = \left[\sum_j \exp\left(y_j^U - y_i^L\right)\right]^{-1}, \quad p_i^U = \left[\sum_j \exp\left(y_j^L - y_i^U\right)\right]^{-1}$$

The categorical cross entropy loss and its first partial derivative are given by

$$\mathcal{L}\left(y^{\star}, y'\right) = -\sum_i y_i^t \log p_i, \quad \frac{\partial \mathcal{L}\left(y^{\star}, y'\right)}{\partial y^{\star}} = p - y^t$$

where $y^t$ is a one-hot encoding of the true label. Considering label flipping attacks ($q = 0, \nu = 1$), we can bound the partial derivative by

$$\left[\partial l^L\right]_i = p_i^L - 1, \quad \left[\partial l^U\right]_i = p_i^U - 0$$

In the case of targeted label flipping attacks (e.g. only applying label flipping attacks to / from specific classes), stronger bounds can be obtained by considering the $0 - 1$ bounds only on the indices $y_i^t$ affected by the attack. The cross entropy loss itself is bounded by $l^L = -\sum_i y_i^t \log p_i^U, l^U = -\sum_i y_i^t \log p_i^L$.

**Interval Back-Propagation.** We extend the interval arithmetic based approach of Wicker et al. (2022), which bounds derivatives of the form $\partial \mathcal{L}/\partial z^{(k)}$, to additionally compute bounds on the derivatives w.r.t. the parameters. First, we back-propagate intervals over $y^\star$ (the label) and $\hat{z}^{(K)}$ (the logits) to compute an interval over $\partial \mathcal{L}/\partial \hat{z}^{(K)}$, the gradient of the loss w.r.t. the logits of the network. The procedure for computing this interval is described in Appendix C for a selection of loss functions. We then use interval bound propagation to back-propagate this interval through the network to compute intervals over all gradients:

$$\frac{\partial \mathcal{L}}{\partial z^{(k-1)}} = \left( W^{(k)} \right)^\top \otimes \frac{\partial \mathcal{L}}{\partial \hat{z}^{(k)}}, \quad \frac{\partial \mathcal{L}}{\partial \hat{z}^{(k)}} = \left[ H\left( \hat{z}_L^{(k)} \right), H\left( \hat{z}_U^{(k)} \right) \right] \odot \frac{\partial \mathcal{L}}{\partial z^{(k)}}$$

$$\frac{\partial \mathcal{L}}{\partial W^{(k)}} = \frac{\partial \mathcal{L}}{\partial \hat{z}^{(k)}} \otimes \left[ \left( z_L^{(k-1)} \right)^\top, \left( z_U^{(k-1)} \right)^\top \right], \quad \frac{\partial \mathcal{L}}{\partial b^{(k)}} = \frac{\partial \mathcal{L}}{\partial \hat{z}^{(k)}}$$

where $H(\cdot)$ is the Heaviside function, and $\circ$ is the element-wise product. The resulting gradient intervals suffice to bound all solutions to our original optimization problem (8). That is, the gradients of the network lie within these intervals for all $W^{(k)} \in W^{(k)}$, $b^{(k)} \in b^{(k)}$, $\|x - x^\star\|_p \leq \epsilon$, and $\|y - y^\star\|_q \leq \nu$.

# D Bounding the Descent Direction for an Unbounded Adversary

**Theorem D.1** (Bounding the descent direction for an unbounded adversary). *Given a nominal batch $\mathcal{B} = \left\{ \left( x^{(i)}, y^{(i)} \right) \right\}_{i=1}^b$ with batchsize $b$, a parameter set $\left[ \theta^L, \theta^U \right]$, and a clipping level $\kappa$, the clipped SGD parameter update $\Delta\theta = \frac{1}{b} \sum_{\widetilde{\mathcal{B}}} \mathrm{Clip}_\kappa \left[ \nabla_\theta \mathcal{L} \left( f^\theta \left( \tilde{x}^{(i)} \right), \tilde{y}^{(i)} \right) \right]$ is bounded element-wise by*

$$\Delta\theta^L = \frac{1}{b} \left( \underset{b-n}{\mathrm{SEMin}} \left\{ \delta_L^{(i)} \right\}_{i=1}^b - n\kappa \mathbf{1}_d \right), \quad \Delta\theta^U = \frac{1}{b} \left( \underset{b-n}{\mathrm{SEMax}} \left\{ \delta_U^{(i)} \right\}_{i=1}^b + n\kappa \mathbf{1}_d \right) \tag{23}$$

*for any poisoned batch $\widetilde{\mathcal{B}}$ derived from $\mathcal{B}$ by substituting up to $n$ data-points with poisoned data and any $\theta \in [\theta^L, \theta^U]$. The terms $\delta_L^{(i)}, \delta_U^{(i)}$ are sound bounds that account for the worst-case effect of additions/removals in any previous iterations. That is, they bound the gradient given any parameter $\theta^\star \in [\theta^L, \theta^U]$ in the reachable set, i.e. for all $i = 1, \ldots, b$, we have $\delta_L^{(i)} \preceq \delta^{(i)} \preceq \delta_U^{(i)}$ for any*

$$\delta^{(i)} \in \left\{ \mathrm{Clip}_\kappa \left[ \nabla_{\theta'} \mathcal{L} \left( f^{\theta'}(x^{(i)}), y^{(i)} \right) \right] \mid \theta' \in \left[ \theta^L, \theta^U \right] \right\}. \tag{24}$$

The operations $\mathrm{SEMax}_a$ and $\mathrm{SEMin}_a$ correspond to taking the sum of the element-wise top/bottom-$a$ elements over each index of the input vectors. Therefore, the update step in (23) corresponds to substituting the $n$ elements with the *largest / smallest* gradients (by taking the sum of only the min / max $b-n$ gradients) with the *minimum / maximum* possible gradient updates ($-\kappa, \kappa$, respectively, due to the clipping operation). Since we wish to soundly over-approximate this operation for all parameters, we perform this bounding operation independently over each index of the parameter vector. This is certainly a loose approximation, as the $n$ points that maximize the gradient at a particular index will likely not maximize the gradient of other indices. Note that without clipping, the min / max effect of adding arbitrary data points into the training data is unbounded and we cannot compute any guarantees.

# E Comparison with Empirical Attacks

In this section, we compare the tightness of our bounds with simple heuristic poisoning attacks for both the UCI-houseelectric and OCT-MNIST datasets.

## E.1 Visualising Attacks in Parameter Space (UCI-houseelectric)

First, we investigate the tightness of our bounds in parameter space via a feature poisoning attack. The attack's objective is to maximize a given scalar function of the parameters, which we label $f^{\mathrm{targ}}(\theta)$. We then take the following poisoning procedure at each training iteration:

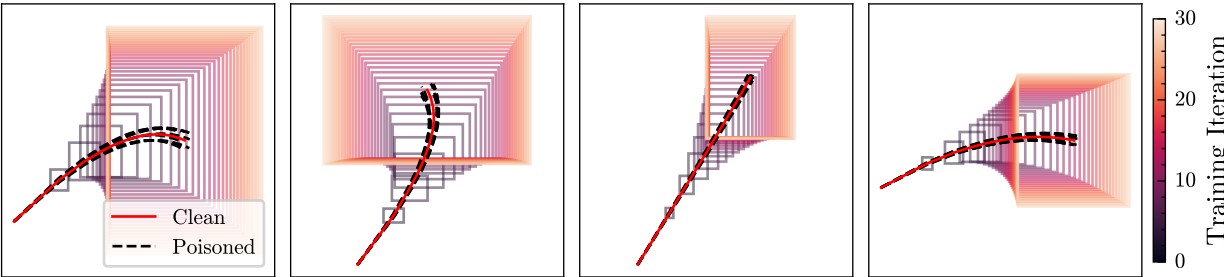

Figure 7: Training trajectory for selected parameters under parameter-targeted feature poisoning with an adversary of $\epsilon = 0.02, n = 2000, p = \infty$. The coloured boxes show the bounds $[\theta_L, \theta_U]$ obtained at each training iteration using AGT.

1. Randomly sample a subset of $n$ samples from the current training batch.

2. For each selected sample $x^{(i)}$, compute a poison $v^{(i)}$ such that $x^{(i)} + v^{(i)}$ maximizes the gradient $\partial f^{\text{targ}}/\partial x$ subject to $\|v^{(i)}\|_p \leq \epsilon$.

3. Add the noise to each of the $n$ sampled points to produce the poisoned dataset.

The noise in step 2 is obtained via projected gradient descent (PGD). To visualise the effect of our attack in parameter space, we plot the trajectory taken by two randomly selected parameters $\theta_i, \theta_j$ from the network. We then run our poisoning attack on a collection of poisoning objectives $f^{\text{targ}}$, such as $\theta_i + \theta_j, \theta_i, -\theta_j$, etc. The effect of our poisoning attack is to perturb the training trajectory in the direction to maximize the given objective.

Figure 7 shows the result of this poisoning procedure for a random selection of parameter training trajectories. We can see that the poisoned trajectories (in black) lie close the clean poisoned trajectory, while our bounds represent an over-approximation of all the possible training trajectories.

### E.2 Feature-Space Collision Attack (UCI-houseelectric)

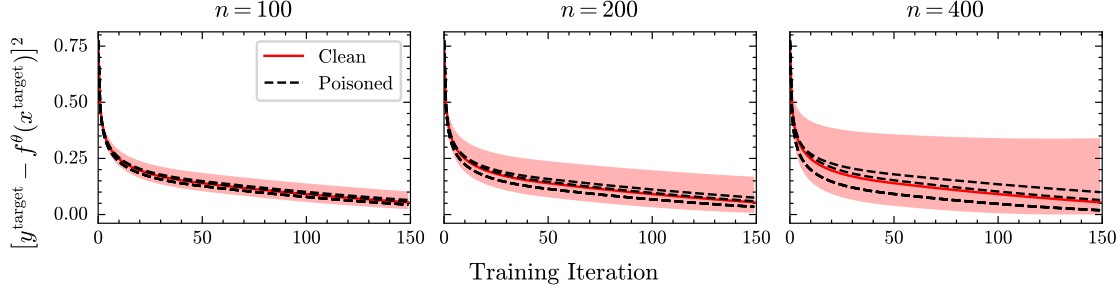

Figure 8: Mean squared error on the target point $(x^{\text{target}}, y^{\text{target}})$ in the UCI-houseelectric dataset. Black lines show loss trajectories under the randomized feature-collision poisoning attack.

We now consider an unbounded attack setting where the adversary's goal is to prevent the model from learning a particular training example $(x^{\text{target}}, y^{\text{target}})$. We again consider a simple randomized attack setting, where the adversary first selects a subset of $n$ samples from each training batch. The adversary then replaces the features of each of the $n$ samples with $x^{\text{target}}$, and assigns each one a randomly generated label. In this way, the adversary aims to obscure the true target label and prevent the model from learning the pair $(x^{\text{target}}, y^{\text{target}})$.

Figure 8 shows the loss of the model on the target point at each training iteration. To investigate the tightness of our loss lower bound, we also consider the case where the adversary replaces all of the $n$ sampled instances from the batch with the true $(x^{\text{target}}, y^{\text{target}})$, thus over-representing the sample within the batch and causing the model to fit the target point faster. The bounds (in red) are obtained from AGT with an unbounded adversary ($\kappa = 0.05$). We can see that although our bounds are not tight to any of the attacks considered, they remain sound for all the poisoned training trajectories.

### E.3 Randomized Label Flipping Attack (OCT-MNIST)

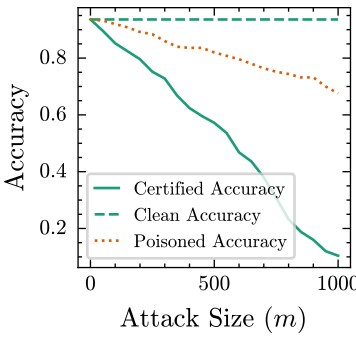

Here, we present the results of a label-flipping attack conducted on the OCT-MNIST dataset. Following the approach described in Section 4, we begin by pre-training a binary classification model to distinguish between two diseased classes and a healthy class. Next, we fine-tune the model's final dense layers on the 'Drusen' class, which we assume to be potentially compromised, using a training set composed of 50% clean data and 50% Drusen data, with 3000 samples from each category in each batch. Given that the Drusen class is a minority, we simulate a scenario where a random subset of the Drusen data is incorrectly labeled as the 'healthy' class. Figure 9 displays the model's accuracy when trained on the poisoned dataset. We can see that training on the mis-labelled data results in a significant decrease in model accuracy, though the poisoned accuracy remains within the bounds of certified by AGT.

Figure 9: Accuracy on the OCT-MNIST dataset under a random label flipping attack on the Drusen class.

## F Proofs

### F.1 Proof of Theorem 3.2 (Bounding Adversary Goals via Parameter Space Bounds)

We begin the proof by writing out the form of the function we wish to optimize, $J$, for each attack setting considered. Below the right hand side of the inequality is taken to be the function $J$, and each inequality is the statement we would like to prove.

For denial of service our bound becomes:

$$\max_{\mathcal{D}' \in \mathcal{T}} \frac{1}{k} \sum_{i=1}^{k} \mathcal{L}\big(f^{M(f,\theta',\mathcal{D}')}(x^{(i)}), y^{(i)}\big) \leq \max_{\theta^\star \in [\theta^L, \theta^U]} \frac{1}{k} \sum_{i=1}^{k} \mathcal{L}\big(f^{\theta^\star}(x^{(i)}), y^{(i)}\big)$$

For certified prediction poisoning robustness our bound becomes:

$$\max_{\mathcal{D}' \in \mathcal{T}} \frac{1}{k} \sum_{i=1}^{k} \mathbb{1}\big(f^{M(f,\theta',\mathcal{D}')}(x^{(i)}) \notin S\big) \leq \max_{\theta^\star \in [\theta^L, \theta^U]} \frac{1}{k} \sum_{i=1}^{k} \mathbb{1}\big(f^{\theta^\star}(x^{(i)}) \notin S\big)$$

And for backdoor attacks our bound becomes:

$$\max_{\mathcal{D}' \in \mathcal{T}} \frac{1}{k} \sum_{i=1}^{k} \mathbb{1}\big(\exists x^\star \in V(x^{(i)}) \ s.t. \ f^{M(f,\theta',\mathcal{D}')}(x^\star) \notin S\big) \leq \max_{\theta^\star \in [\theta^L, \theta^U]} \frac{1}{k} \sum_{i=1}^{k} \mathbb{1}\big(\exists x^\star \in V(x^{(i)}) \ s.t. \ f^{\theta^\star}(x^\star) \notin S\big)$$

*Proof.* Without loss of generality, take the function we wish to optimize to be denoted simply by $J$. By definition, there exists a parameter, $\theta^\circ = M(f, \theta', \mathcal{D}')$ resulting from a particular dataset $\mathcal{D}' \in \mathcal{T}(\mathcal{D})$ such that $\theta^\circ$ provides a (potentially non-unique) optimal solution to the optimization problem we wish to bound, i.e., the left hand side of the inequalities above. Given a valid parameter space bound $[\theta^L, \theta^U]$ satisfying Equation 1, we have that necessarily, $\theta^\circ \in [\theta^L, \theta^U]$. Therefore, the result of optimizing over $[\theta^L, \theta^U]$ can provide at a minimum the bound realized by $\theta^\circ$; however, due to approximation, this bound might not be tight, so optimizing over $[\theta^L, \theta^U]$ provides an upper-bound, thus proving the inequalities above. □

## F.2 Proof of Theorem 3.3 (Algorithm Correctness)

*Proof.* Here we provide a proof of correctness for our algorithm (i.e., proof of Theorem 3.3) as well as a detailed discussion of the operations therein.

First, we recall the definition of valid parameter space bounds (Equation 1 in the main text):

$$\theta_i^L \leq \min_{\mathcal{D}' \in \mathcal{T}(\mathcal{D})} M(f, \theta', \mathcal{D}')_i \leq M(f, \theta', \mathcal{D})_i \leq \max_{\mathcal{D}' \in \mathcal{T}(\mathcal{D})} M(f, \theta', \mathcal{D}')_i \leq \theta_i^U$$

As well as the iterative equations for stochastic gradient descent:

$$\theta \leftarrow \theta - \alpha \Delta\theta, \qquad \Delta\theta \leftarrow \frac{1}{|\mathcal{B}|} \sum_{(x,y) \in \mathcal{B}} \nabla_\theta \mathcal{L}\left(f^\theta(x), y\right)$$

For ease of notation, we assume a fixed data ordering (one may always take the element-wise maximums/minimums over the entire dataset rather than each batch to relax this assumption).

Now, we proceed to prove by induction that Algorithm 1 maintains valid parameter space bounds on each step of gradient descent. We start with the base case of $\theta^L = \theta^U = \theta'$ according to line 1, which are valid parameter-space bounds. Our inductive hypothesis is that, given valid parameter space bounds satisfying Definition 3.1, each iteration of Algorithm 1 (lines 4–8) produces a new $\theta^L$ and $\theta^U$ that satisfy also Definition 3.1.

First, we observe that lines 4–5 simply compute the normal forward pass. Second, we note that lines 6–7 compute valid bounds on the descent direction for all possible poisoning attacks within $\mathcal{T}(\mathcal{D})$. In other words, the inequality $\Delta\theta^L \leq \Delta\theta \leq \Delta\theta^U$ holds element-wise for any possible batch $\tilde{\mathcal{B}} \in \mathcal{T}(\mathcal{D})$. Combining this largest and smallest possible update with the smallest and largest previous parameters yields the following bounds:

$$\theta^L - \alpha \Delta\theta^U \leq \theta - \alpha \Delta\theta \leq \theta^U - \alpha \Delta\theta^L$$

which, by definition, constitute valid parameter-space bounds and, given that these bounds are exactly those in Algorithm 1, we have that Algorithm 1 provides valid parameter space bounds as desired. $\square$

## F.3 Proof of Theorem D.1 (Descent Direction Bound for Unbounded Adversaries)

The nominal clipped descent direction for a parameter $\theta$ is the averaged, clipped gradient over a training batch $\mathcal{B}$, defined as

$$\Delta\theta = \frac{1}{b} \sum_{i=1}^{b} \text{Clip}_\kappa \left[\delta^{(i)}\right]$$

where each gradient term is given by $\delta^{(i)} = \nabla_\theta \mathcal{L}\left(f^\theta\left(x^{(i)}\right), y^{(i)}\right)$. Our goal is to bound this descent direction for the case when (up to) $n$ points are removed or added to the training data, for any $\theta \in [\theta_L, \theta_U]$. We begin by bounding the descent direction for a fixed, scalar $\theta$, then generalize to all $\theta \in [\theta_L, \theta_U]$ and to the multi-dimensional case (i.e., multiple parameters). We present only the upper bounds here; analogous results apply for lower bounds.

**Bounding the descent direction for a fixed, scalar $\theta$.** Consider the effect of removing up to $n$ data points from batch $\mathcal{B}$. Without loss of generality, assume the gradient terms are sorted in descending order, i.e., $\delta^{(1)} \geq \delta^{(2)} \geq \cdots \geq \delta^{(b)}$. Then, the average clipped gradient over all points can be bounded above by the average over the largest $b - n$ terms:

$$\Delta\theta = \frac{1}{b} \sum_{i=1}^{b} \text{Clip}_\kappa \left[\delta^{(i)}\right] \leq \frac{1}{b-n} \sum_{i=1}^{b-n} \text{Clip}_\kappa \left[\delta^{(i)}\right]$$

This bound corresponds to removing the $n$ points with the smallest gradients.

Next, consider adding $n$ arbitrary points to the training batch. Since each added point contributes at most $\kappa$ due to clipping, the descent direction with up to $n$ removals and $n$ additions is bounded by

$$\frac{1}{b}\sum_{i=1}^{b}\mathrm{Clip}_\kappa\left[\delta^{(i)}\right] \leq \frac{1}{b-n}\sum_{i=1}^{b-n}\mathrm{Clip}_\kappa\left[\delta^{(i)}\right] \leq \frac{1}{b}\left(n\kappa + \sum_{i=1}^{b}\mathrm{Clip}_\kappa\left[\delta^{(i)}\right]\right)$$

where the bound now accounts for replacing the $n$ smallest gradient terms with the maximum possible value of $\kappa$ from the added samples.

**Bounding the effect of a variable parameter interval.** We extend this bound to any $\theta \in [\theta_L, \theta_U]$. Assume the existence of upper bounds $\delta_U^{(i)}$ on the clipped gradients for each data point over the interval, such that

$$\delta_U^{(i)} \geq \mathrm{Clip}_\kappa\left[\nabla_{\theta'}\mathcal{L}\left(f^{\theta'}(x^{(i)}), y^{(i)}\right)\right] \quad \forall \theta' \in [\theta_L, \theta_U].$$

Then, using these upper bounds, we further bound $\Delta\theta$ as

$$\Delta\theta \leq \frac{1}{b}\left(n\kappa + \sum_{i=1}^{b}\mathrm{Clip}_\kappa\left[\delta_U^{(i)}\right]\right)$$

where, as before, we assume $\delta_U^{(i)}$ are indexed in descending order.

**Extending to the multi-dimensional case.** To generalize to the multi-dimensional case, we apply the above bound component-wise. Since gradients are not necessarily ordered for each parameter component, we introduce the $\mathrm{SEMax}_n$ operator, which selects and sums the largest $n$ terms at each index. This yields the following bound on the descent direction:

$$\Delta\theta \leq \frac{1}{b}\left(\mathop{\mathrm{SEMax}}_{b-n}\left\{\delta_U^{(i)}\right\}_{i=1}^{b} + n\kappa\mathbf{1}_d\right)$$

which holds for any $\theta \in [\theta_L, \theta_U]$ and up to $n$ removed and replaced points. $\qquad\square$

We have established the upper bound on the descent direction. The corresponding lower bound can be derived by reversing the inequalities and substituting SEMax with the analagous minimization operator, SEMin.

### F.4 Proof of Theorem 3.4 (Descent Direction Bound for Bounded Adversaries)

The nominal descent direction for a parameter $\theta$ is the averaged gradient over a training batch $\mathcal{B}$, defined as

$$\Delta\theta = \frac{1}{b}\sum_{i=1}^{b}\delta^{(i)}$$

where each gradient term is given by $\delta^{(i)} = \nabla_\theta\mathcal{L}\left(f^\theta\left(x^{(i)}\right), y^{(i)}\right)$. Our goal is to upper bound this descent direction when up to $n$ points from $\mathcal{B}$ are poisoned by up to $\epsilon$ in the feature space and $\nu$ in label space. The bound is additionally computed with respect to any $\theta \in [\theta_L, \theta_U]$. We again begin by bounding the descent direction for a fixed $\theta$, then generalize to all $\theta \in [\theta_L, \theta_U]$. We present only the upper bounds here, though corresponding results for the lower bound can be shown by reversing the inequalities and replacing SEMax with SEMin.

**Bounding the descent direction for a fixed $\theta$.** Consider the effect of poisoning either the features or the labels of a data point. For a given data point, an adversary may choose to poison its features, its labels, or both. In total, at most $n$ points may be influenced by the poisoning adversary. We assume that $n \leq b$, otherwise take at most $\min(n, b)$ points to be poisoned.

Assume that we have access to sound gradient upper bounds

$$\delta^{(i)} \leq \tilde{\delta}_U^{(i)} \quad \forall \delta^{(i)} \in \left\{\nabla_{\theta'}\mathcal{L}\left(f^{\theta'}(\tilde{x}), \tilde{y}\right) \mid \|x^{(i)} - \tilde{x}\|_p \leq \epsilon, \|y^{(i)} - \tilde{y}\|_q \leq \nu\right\}.$$

where the inequalities are interpreted element-wise. Here, $\tilde{\delta}_U^{(i)}$ corresponds to an upper bound on the maximum possible gradient achievable at the data-point $(x^{(i)}, y^{(i)})$ through poisoning.

The adversary's maximum possible impact on the descent direction at any point $i$ is given by $\tilde{\delta}_U^{(i)} - \delta^{(i)}$. To maximize an upper bound on $\Delta\theta$, we consider the $n$ points with the largest possible adversarial contributions. Therefore, we obtain

$$\Delta\theta = \frac{1}{b}\sum_{i=1}^{b}\delta^{(i)} \leq \frac{1}{b}\left(\underset{n}{\text{SEMax}}\left\{\tilde{\delta}_U^{(i)} - \delta^{(i)}\right\}_{i=1}^{b} + \sum_{i=1}^{b}\delta^{(i)}\right),$$

where the SEMax operation corresponds to taking the sum of the largest $n$ elements of its argument at each element. This bound captures the maximum increase in $\Delta\theta$ that an adversary can induce by poisoning up to $n$ data points.

**Bounding the effect of a variable parameter interval.** Now, we wish to compute a bound on $\Delta\theta$ for any $\theta \in [\theta_L, \theta_U]$. To achieve this, we extend our previous gradient bounds to account for the interval over our parameters. Specifically, we define upper bounds on the nominal and adversarially perturbed gradients that hold across the entire parameter interval:

$$\delta \leq \delta_U^{(i)} \quad \forall \delta \in \left\{\nabla_{\theta'}\mathcal{L}\left(f^{\theta'}(x^{(i)}), y^{(i)}\right) \mid \theta' \in [\theta^L, \theta^U]\right\},$$

$$\tilde{\delta} \leq \tilde{\delta}_U^{(i)} \quad \forall \tilde{\delta} \in \left\{\nabla_{\theta'}\mathcal{L}\left(f^{\theta'}(\tilde{x}), \tilde{y}\right) \mid \theta' \in [\theta^L, \theta^U], \|x^{(i)} - \tilde{x}\|_p \leq \epsilon, \|y^{(i)} - \tilde{y}\|_q \leq \nu\right\}.$$

Thus, the descent direction is upper bounded by

$$\Delta\theta \leq \Delta\theta^U = \frac{1}{b}\left(\underset{n}{\text{SEMax}}\left\{\tilde{\delta}_U^{(i)} - \delta_U^{(i)}\right\}_{i=1}^{b} + \sum_{i=1}^{b}\delta_U^{(i)}\right)$$

for all $\theta \in [\theta_L, \theta_U]$, where the appropriate bounds with respect to the parameter interval have been substituted in.

## F.5 Proof of Proposition 3.5 (CROWN bounds)

To prove Proposition 3.5, we rely on the following result which we reproduce from Zhang et al. (2018):

**Theorem F.1** (Explicit output bounds of a neural network $f$ (Zhang et al., 2018))**.** *Given an $m$-layer neural network function $f : \mathbb{R}^{n_0} \to \mathbb{R}^{n_m}$, there exists two explicit functions $f_j^L : \mathbb{R}^{n_0} \to \mathbb{R}$ and $f_j^U : \mathbb{R}^{n_0} \to \mathbb{R}$ such that $\forall j \in [n_m], \forall x \in \mathbb{B}_p(x_0, \epsilon)$, the inequality $f_j^L(x) \leq f_j(x) \leq f_j^U(x)$ holds true, where*

$$f_j^U(x) = \Lambda_{j,:}^{(0)}x + \sum_{k=1}^{m}\Lambda_{j,:}^{(k)}\left(b^{(k)} + \Delta_{:,j}^{(k)}\right), \quad \Lambda_{j,:}^{(k-1)} = \begin{cases} e_j^\top & \text{if } k = m+1; \\ \left(\Lambda_{j,:}^{(k)}W^{(k)}\right) \circ \lambda_{j,:}^{(k-1)} & \text{if } k \in [m]. \end{cases}$$

$$f_j^L(x) = \Omega_{j,:}^{(0)}x + \sum_{k=1}^{m}\Omega_{j,:}^{(k)}\left(b^{(k)} + \Theta_{:,j}^{(k)}\right), \quad \Omega_{j,:}^{(k-1)} = \begin{cases} e_j^\top & \text{if } k = m+1; \\ \left(\Omega_{j,:}^{(k)}W^{(k)}\right) \circ \omega_{j,:}^{(k-1)} & \text{if } k \in [m] \end{cases}$$

*and $\forall i \in [n_k]$, we define four matrices $\lambda^{(k)}, \omega^{(k)}, \Delta^{(k)}, \Theta^{(k)} \in \mathbb{R}^{n_m \times n_k}$ :*

$$\lambda_{j,i}^{(k)} = \begin{cases} \alpha_{U,i}^{(k)} & \text{if } k \neq 0, \Lambda_{j,:}^{(k+1)}W_{:,i}^{(k+1)} \geq 0; \\ \alpha_{L,i}^{(k)} & \text{if } k \neq 0, \Lambda_{j,:}^{(k+1)}W_{:,i}^{(k+1)} < 0; \\ 1 & \text{if } k = 0. \end{cases} \quad \omega_{j,i}^{(k)} = \begin{cases} \alpha_{L,i}^{(k)} & \text{if } k \neq 0, \Omega_{j,:}^{(k+1)}W_{:,i}^{(k+1)} \geq 0; \\ \alpha_{U,i}^{(k)} & \text{if } k \neq 0, \Omega_{j,:}^{(k+1)}W_{:,i}^{(k+1)} < 0; \\ 1 & \text{if } k = 0. \end{cases}$$

$$\Delta_{i,j}^{(k)} = \begin{cases} \beta_{U,i}^{(k)} & \text{if } k \neq m, \Lambda_{j,:}^{(k+1)}W_{:,i}^{(k+1)} \geq 0; \\ \beta_{L,i}^{(k)} & \text{if } k \neq m, \Lambda_{j,:}^{(k+1)}W_{:,i}^{(k+1)} < 0; \\ 0 & \text{if } k = m. \end{cases} \quad \Theta_{i,j}^{(k)} = \begin{cases} \beta_{L,i}^{(k)} & \text{if } k \neq m, \Omega_{j,:}^{(k+1)}W_{:,i}^{(k+1)} \geq 0; \\ \beta_{U,i}^{(k)} & \text{if } k \neq m, \Omega_{j,:}^{(k+1)}W_{:,i}^{(k+1)} < 0; \\ 0 & \text{if } k = m. \end{cases}$$

*and $\circ$ is the Hadamard product and $e_j \in \mathbb{R}^{n_m}$ is a standard unit vector at $j$ th coordinate.*

The terms $\alpha_{L,i}^{(k)}, \alpha_{U,i}^{(k)}, \beta_{L,i}^{(k)}$, and $\beta_{U,i}^{(k)}$ represent the coefficients of linear bounds on the activation functions, that is for the $r$-th neuron in $k$-th layer with activation function $\sigma(x)$, there exist two linear functions

$$h_{L,r}^{(k)}(x) = \alpha_{L,r}^{(k)}\left(x + \beta_{L,r}^{(k)}\right), \quad h_{U,r}^{(k)}(x) = \alpha_{U,r}^{(k)}\left(x + \beta_{U,r}^{(k)}\right)$$

such that $h_{L,r}^{(k)}(x) \leq \sigma(x) \leq h_{U,r}^{(k)}(x) \; \forall x \in \left[l_r^{(k)}, u_r^{(k)}\right]$. The terms $\left[l_r^{(k)}, u_r^{(k)}\right]$ are assumed to be sound bounds on all previous neurons in the network. We first note that, for any neuron $r$ in the $k$-th layer, the linear bounds may be replaced with concrete bounds by substituting $\alpha_{L,r}^{(k)} = \alpha_{U,r}^{(k)} = 0$ and $\beta_{L,r}^{(k)} = l_r^{(k)}, \beta_{U,r}^{(k)} = u_r^{(k)}$. Let $S^L, S^U$ be index sets of tuples $(i,k)$ indicating whether the lower and upper bounds (respectively) of the $i$-th neuron in the $k$-th layer should be concretized in this way. Then, the equivalent weights and biases take the following form:

$$\lambda_{j,i}^{(k)} = \begin{cases} \alpha_{U,i}^{(k)} & \text{if } (i,k) \notin S^U, k \neq 0, \Lambda_{j,:}^{(k+1)}W_{:,i}^{(k+1)} \geq 0; \\ \alpha_{L,i}^{(k)} & \text{if } (i,k) \notin S^U, k \neq 0, \Lambda_{j,:}^{(k+1)}W_{:,i}^{(k+1)} < 0; \\ 1 & \text{if } (i,k) \notin S^U, k = 0; \\ 0 & \text{if } (i,k) \in S^U. \end{cases}$$

$$\omega_{j,i}^{(k)} = \begin{cases} \alpha_{L,i}^{(k)} & \text{if } (i,k) \notin S^L, k \neq 0, \Omega_{j,:}^{(k+1)}W_{:,i}^{(k+1)} \geq 0; \\ \alpha_{U,i}^{(k)} & \text{if } (i,k) \notin S^L, k \neq 0, \Omega_{j,:}^{(k+1)}W_{:,i}^{(k+1)} < 0; \\ 1 & \text{if } (i,k) \notin S^L, k = 0; \\ 0 & \text{if } (i,k) \in S^L. \end{cases}$$

$$\Delta_{i,j}^{(k)} = \begin{cases} \beta_{U,i}^{(k)} & \text{if } (i,k) \notin S^U, k \neq m, \Lambda_{j,:}^{(k+1)}W_{:,i}^{(k+1)} \geq 0; \\ \beta_{L,i}^{(k)} & \text{if } (i,k) \notin S^U, k \neq m, \Lambda_{j,:}^{(k+1)}W_{:,i}^{(k+1)} < 0; \\ 0 & \text{if } (i,k) \notin S^U, k = m; \\ u^{(k)} & \text{if}(i,k) \in S^U. \end{cases}$$

$$\Theta_{i,j}^{(k)} = \begin{cases} \beta_{L,i}^{(k)} & \text{if } (i,k) \notin S^L, k \neq m, \Omega_{j,:}^{(k+1)}W_{:,i}^{(k+1)} \geq 0; \\ \beta_{U,i}^{(k)} & \text{if } (i,k) \notin S^L, k \neq m, \Omega_{j,:}^{(k+1)}W_{:,i}^{(k+1)} < 0; \\ 0 & \text{if } (i,k) \notin S^L, k = m; \\ l^{(k)} & \text{if}(i,k) \in S^L. \end{cases}$$

This is exactly the form described in Appendix C, where $S^L$ and $S^U$ are chosen to be the sets of neurons whose equivalent coefficient interval spans zero. Without this modification, the equivalent weights and biases of such neurons in the original formulation would be undefined. We now have bounds on the output of the neural network for which all operations are well-defined in interval arithmetic.

Replacing all operations in the computation of the equivalent weight terms by their interval arithmetic counterparts, we can compute sound, though over-approximated, intervals over $\Lambda^{(0:m)}$ and $\Omega^{(0:m)}$ which satisfy

$$\Lambda^{(k)} \in \boldsymbol{\Lambda^{(k)}} \quad \forall \, W^{(1:m)} \in \boldsymbol{W}^{(1:m)}, b^{(1:m)} \in \boldsymbol{b}^{(1:m)},$$
$$\Omega^{(k)} \in \boldsymbol{\Omega^{(k)}} \quad \forall \, W^{(1:m)} \in \boldsymbol{W}^{(1:m)}, b^{(1:m)} \in \boldsymbol{b}^{(1:m)}.$$

This is trivially true by the definitions of the interval arithmetic operations as given in Appendix C.

Turning to the upper bound (though analogous arguments hold for the lower bound), we have

$$f_j^U\left(x, \Lambda^{(0:m)}, \Delta^{(1:m)}, b^{(1:m)}\right) = \Lambda_{j,:}^{(0)}x + \sum_{k=1}^{m}\Lambda_{j,:}^{(k)}\left(b^{(k)} + \Delta_{:,j}^{(k)}\right)$$

where $\Lambda^{(0:m)}$ are functions of the weights and biases of the network and $\Delta^{(1:m)}$ are constants that depend on the bounds on the intermediate layers of the network. Thus, given parameter intervals $\boldsymbol{b}^{(k)}, \boldsymbol{W}^{(k)}$, the

following result holds

$$f_j^U\left(x, \Lambda^{(0:m)}, \Delta^{(1:m)}, b^{(1:m)}\right) \leq \max\left\{ f_j^U\left(x, \Lambda^{(0:m)}, \Delta^{(1:m)}, b^{(1:m)}\right) \mid \begin{array}{c} W^{(1:m)} \in \boldsymbol{W}^{(1:m)} \\ b^{(1:m)} \in \boldsymbol{b}^{(1:m)} \end{array} \right\}$$

$$\leq \max\left\{ f_j^U\left(x, \Lambda^{(0:m)}, \Delta^{(1:m)}, b^{(1:m)}\right) \mid \begin{array}{c} \Lambda^{(0:m)} \in \boldsymbol{\Lambda}^{(0:m)} \\ b^{(1:m)} \in \boldsymbol{b}^{(1:m)} \end{array} \right\}$$

for any set of intervals $\boldsymbol{\Lambda}^{(0:m)}$ that satisfy $\{\Lambda^{(k)} \mid W^{(1:m)} \in \boldsymbol{W}^{(1:m)}\} \subseteq \boldsymbol{\Lambda}^{(k)}$. Since our intervals $\boldsymbol{\Lambda}^{(0:m)}$ computed via interval arithmetic satisfy this property, we have that any valid bound on this maximization problem constitutes a bound on the output of the neural network $f_j(x)$ for any $W^{(1:m)} \in \boldsymbol{W}^{(1:m)}$ and $b^{(1:m)} \in \boldsymbol{b}^{(1:m)}$.

## G  Experimental Details and Runtimes

This section details the datasets and hyper-parameters used for the experiments detailed in Section 4. All experiments were run on a server equipped with 2x AMD EPYC 9334 CPUs and 2x NVIDIA L40 GPUs using an implementation of Algorithm 1 written in Python using Pytorch.

Table 1 shows a run-time comparison of our implementation of Algorithm 1 with (un-certified) training in Pytorch. We observe that training using Abstract Gradient Training typically incurs a modest additional cost per iteration when compared to standard training.

| Dataset | Time per iteration (seconds) | |
| --- | --- | --- |
| | Abstract gradient training | Un-certified training |
| UCI House-electric | 0.25 | 0.12 |
| MNIST (inc. PCA projection) | 1.6 | 1.1 |
| OCT-MNIST | 0.96 | 0.10 |
| Udacity Self-Driving | 53 | 42 |

Table 1: Comparison of the run-time of AGT and standard model training in Pytorch.

Table 2 details the datasets along with the number of epochs, learning rate ($\alpha$), decay rate ($\eta$) and batch size ($b$) used for each. We note that a standard learning rate decay of the form ($\alpha_n = \alpha/(1+\eta n)$) was applied during training. In the case of fine-tuning both OCT-MNIST and PilotNet, each batch consisted of a mix 70% 'clean' data previously seen during pre-training and 30% new, potentially poisoned, fine-tuning data.

| Dataset | #Samples | #Features | #Epochs | $\alpha$ | $\eta$ | $b$ |
| --- | --- | --- | --- | --- | --- | --- |
| UCI House-electric | 2049280 | 11 | 1 | 0.02 | 0.2 | 10000 |
| MNIST | 60000 | 784 | 3 | 5.0 | 1.0 | 60000 |
| OCT-MNIST | 97477 | 784 | 2 | 0.05 | 5.0 | 6000 |
| Udacity Self-Driving | 31573 | 39600 | 2 | 0.25 | 10.0 | 10000 |

Table 2: Datasets and Hyperparameter Settings

