# OpenReview forum: "Certified Robustness to Data Poisoning in Gradient-Based Training"
_TMLR — Accepted by TMLR_

### Review · Reviewer_4DXp · 2025-06-08

**Summary Of Contributions:**

The paper presents a novel framework designed to provide certified robustness guarantees against data poisoning in gradient-based learning algorithms. The proposed framework does not modify the underlying learning algorithm or model; instead, it leverages convex relaxations and bound-propagation techniques to define and track intervals of reachable parameters under various adversarial manipulations (both bounded and unbounded). This allows for certifying robustness in scenarios involving untargeted, targeted, and backdoor attacks, offering theoretical guarantees on the worst-case model performance and behavior under poisoning attacks

**Audience:**

Yes

**Broader Impact Concerns:**

No concerns

**Claims And Evidence:**

Yes

**Requested Changes:**

Please see weaknesses, but mainly clarify the motivation and practical usefulness for parameter intervals, and improve readability of the theoretical structure.

**Strengths And Weaknesses:**

**Strengths:**

- The problem is interesting and the concept of certifying robustness without modifying the underlying learning algorithm is promising, providing generality and flexibility.

- The approach tackles the challenge by trying to bound reachable parameter sets, a novel perspective compared to existing art.

- Consideration of a variety of threat models (bounded and unbounded poisoning attacks) and attack goals (untargeted, targeted, backdoor attacks).

**Weaknesses:**

- The motivation and practicality for **bounding the model parameters** (i.e., maintaining an interval over θ) is not clearly established in the paper. I am not sure why one would track such bounds, especially since θ is not guaranteed to correlate with functional behaviour. More importantly, while in an adversarial robustness case (input manipulation at test time) the bounds on input perturbations make sense practically, in the poisoning case, supposing bounds on parameters does not seem neither intuitive, nor practically useful.
- Besides, expectedly, the empirical results suggest that certified bounds weaken for deep models. I am really unsure about how practically useful these bounds would be, even though I have no fundamental concerns theoretically.
- There is no clear statement of the loop invariant that after each iteration the parameter box [θ_L, θ_U] contains all parameters reachable under the adversary model. While it might be moinor, the mention of the “optional” clipping parameter κ is confusing—it is actually necessary for unbounded attacks and irrelevant otherwise. Without this clarification, it can confuse the reader into thinking the clipping is an unclear artifact.
- These equations use symbols like δ_L and δ_U without clearly stating where they come from. This is confusing, especially since deriving such bounds is one of the main goals of the paper. Only later (Section 3.4 and Appendix C) do we learn that these are obtained using interval-CROWN and IBP. Earlier equations should reference this explicitly. Either change the structure, or mention this explicitly; it got me confused dropping these early without explanation. (similar comment on some proofs, including those in Appendix F (Theorem 3) and Theorem 1 (Appendix E), rely on assumptions or constructs defined in later sections or appendices, without making this dependency clear. This forces the reader to hunt around the paper to piece things together)

---

> ### Author Response · Authors · 2025-07-01
> **Response to Reviewer 4DXp**
>
> We thank the reviewer for their consideration of our paper and address their concerns in our response below.
>
> - The reviewer correctly points out that the main goal of any poisoning certification is to bound the behavior of the trained model in function space, rather than parameter space. We highlight to the reviewer that we bound changes in function space at inference time by propagating our parameter bounds. The key idea is that, for parameterized models, poisoned data must affect the model's final behavior by modifying its parameters during training; parameter space bounds offer a concrete way to quantify and bound the impact of the poisoned data. Identifying the exact influence of poisoned data on the functional behavior of the model would require our parameter space bounds to be exact (e.g., no approximation from assuming intervals etc.) and our inference time propagation to be exact, this is unfortunately computationally infeasible for even relatively small models.
> - Regarding scalability, our intention is to introduce a general framework  in the direction of extending existing methods for inference-time verification of neural networks to the problem of training time certification. In particular, our method is orthogonal to many other approaches in data poisoning, including ensemble-based methods such as DPA. Future works may consider improvements of parts of the framework, e.g., the application of AGT to the ensemble setting, and the adaptation of other verification techniques, such as those based on linear and convex optimization, to the data poisoning setting.
> - We highlight to the reviewer that the loop invariant maintained by our approach is stated in Theorem 3.2. While not stated as such, we prove that for any run of AGT returns a valid parameter space bound. This theorem is then proven by induction in Appendix I.
> - We thank the reviewer for pointing out a lack of clarity in the methodology sections of our work. We have modified our manuscript to provide a clearer introduction of all symbols and concepts in our theoretical exposition. We have additionally moved some bound propagation details from the appendix into the main text to improve the readability of Section 3.

---

### Review · Reviewer_YK3j · 2025-06-13

**Summary Of Contributions:**

This paper studies the certified robustness of the neural networks by providing a framework to compute the valid parameter-space bounds $[\theta_L, \theta_U]$ and such that the all possible model parameters after training in this range, and optimizing in this range could provide the certified robustness of the model, though this may be potentially loose. This is because this estimation accumulates across steps and accumulates by element-wise summarization, and in practice the attacker may not have such ability to craft such poisoned data under the constraints.

To compute the bound $[\theta_L, \theta_U]$ at each gradient updates, this work provides an instantiation of CROWN (Zhang et al. 2018). As CROWN approximate the neural network by linearity approximation with upper bound and lower bound,  $\theta_L$ could be tracked via the lower bound and $\theta_U$ could be tracked via the upper bound.

This work also provides experiments to understand the certified performance of the proposed framework under different settings, feature poisoning, label flipping, feature+label poisoning.

**Audience:**

Yes

**Claims And Evidence:**

No

**Requested Changes:**

Please address my concerns in weaknesses.

**Strengths And Weaknesses:**

## Strengths
1. This work provides a new framework for certified robustness by tracking the model parameters range during training. This framework is general and can be applicable for a wide range of poison attacks such as bounded attacks and unbounded attacks, untargeted and targeted poisoning, and backdoor attacks.

2. Experiment results provide useful discussion to understand the framework performance and comparison to previous works.

## Weaknesses
1. Scalability. The bounds may be too loose as derived via relaxation in deeper networks or more epochs. This is already acknowledged  and discussed as limitations in this work on page 8.

a. scalability in model parameters. I wonder for the MNIST result, is the usage of PCA also due to this? How is the performance if without PCA?

2. Assumption limitations.

a. If I understand correctly, this work assumes the input dataset to the framework to be clean instead of being poisoned, and then try to bound the model parameters space by solving an optimization problem allowing at most n samples to be corrupted at each gradient update. It seems to me that there is gap to this scenario and the motivated scenario where the input dataset is already poisoned like Zhu et al. 2019. If I understand correctly, other works like DPA could still provide the certified robustness to this motivated scenario.

b. data orderings.  The first paragraph of Section 2 states that this framework assumes unrestricted access to data ordering. I think this work do assume the data ordering by assume all poisoned samples in a batch or poisoned samples are concentrated in a few batches rather than uniformly distributed across the batches. Otherwise, I would expect the derived certified robustness to be more loose.

3. Clarity.

a. This work instantiates a extension of the CROWN algorithm to the interval-parameter setting. And in page 7 for the comparison to interval bound, the numerical experiments compute both the CROWN bounds and IBP bounds and take the element-wise tightest bound. It would be better to include the ablation results to help understand the qualitative effect of the proposed extension.

b, page 7 says "at most $n$ poisoned points per batch, rather than per datasets". Does this mean that there could be more than $n$ poisoned points per dataset in the presented experimental results?

---

> ### Author Response · Authors · 2025-07-01
> **Response to Reviewer YK3j**
>
> We thank the reviewer for their detailed response and provide clarifications on all points below.
>
> 1a) As noted in the paper, our method performs particularly poorly on multi-class classification problems when compared with binary classification or regression settings. Therefore, in order to obtain a non-vacuous bound for the MNIST dataset it is necessary to first perform a dimensionality reduction via PCA. While our aim is to introduce a general training-time certification framework, future works may investigate tailored methods for tighter bounds in these settings. Nevertheless, we emphasise that there are already many other settings where dimensionality reduction is not necessary to obtain meaningful guarantees.
>
> 2a) Our framework can be interpreted either by assuming the current dataset is poisoned, or that the current dataset is the clean dataset. Since the distance between the poisoned and clean datasets is symmetric, both the clean and poisoned models must be contained within the parameter interval $[\theta_L, \theta_U]$ returned by our algorithm. When considering point-wise certificates, for example, we can certify whether a prediction is constant for all models in the parameter domain. This certificate guarantees that the behaviour of the clean and poisoned model is the same on a particular query, regardless of whether the training dataset is considered to be clean or poisoned.
>
> 2b) Our framework can provide certified guarantees for a fixed data ordering without making assumptions about how poisoned samples are distributed across batches. The distinction is that previous works such as DPA express their guarantees in terms of poisoned datapoints per dataset. In order to strictly convert our $n$ poisoned points per batch to a guarantee over the whole dataset, one must consider the worst-case of all $n$ points being in a single batch, which does significantly reduce the strength of the guarantees.
>
> However, in practical settings where we can reasonably assume that the adversary does not have access to the data ordering and that poisoned data is uniformly distributed across batches, one can report a "safe" proportion of poisoned data per batch by applying appropriate statistical bounds on the probability of exceeding $n$ poisoned samples per batch. This approach would yield less conservative (but probabilistic) guarantees compared to our worst-case deterministic bounds.
>
> 3a) We agree that including an ablation of the tightness of IBP vs our CROWN bounds would be an important addition to our work. We have updated our experimental section to include results to that effect.
>
> 3b) As mentioned above, if one assumes a uniform distribution of poisoned data per batch our method indeed gives guarantees for up to $n$ times the number of batches poisoned points. However, if no assumptions on the distribution of poisoned data are made our guarantees apply only to $n$ points over the entire dataset (to account for the worst-case of placing all $n$ points in a single batch).

---

> > ### Comment · Reviewer_YK3j · 2025-07-13
> > **Thanks for your reply**
> >
> > Thank you for providing detailed response and updated experiments in manuscript! This that help clarified a few points.  I have the following follow-up questions.
> >
> >
> > > clean dataset assumption. "Since the distance between the poisoned and clean datasets is symmetric."
> >
> > While it is the true the distance between the poisoned and clean datasets is symmetric, base on my understanding of the method, this work does not assume pre-requirement for the data. For example, if there is already $k$ (k is smaller than the batch size) data points poisoned, the algorithm will randomly sample $n$ points to derive the bound, the sample points could be overlapped with those already poisoned points, or no overlapped at all, therefore the poisoned points in analysis could be in the range of [max(k, n), min(b, k+n)].
> >
> >
> > > data ordering assumptions.  "worst-case of all points being in a single batch."
> >
> > It is not straightforward to me that the worst-case for $n$ points as the training dynamics of neural networks is complicated. I wonder if the authors could elaborate this a bit or provide some literatures.
> >
> > > dimensionality reduction in MNIST.
> >
> > It would be helpful to discuss the scalability of this method from the model parameters perspective.

---

> > > ### Author Response · Authors · 2025-07-23
> > > **Response to Reviewer YK3j**
> > >
> > > Thank you for the follow-up questions, we provide clarifications below:
> > >
> > > *Clean dataset assumption:*
> > > Note that our method makes no assumption on sampling $n$ points per batch. Instead, our method soundly considers the worst-case of any $n$ points in the batch being poisoned. Thus, if $k\leq n$, our bounds soundly account for the case that $k$ points were already poisoned. Therefore, in the case that the nominal training dataset is the poisoned one, the final parameter-space bounds are guaranteed to contain models trained on both the poisoned and the clean datasets. On the other hand, if $k > n$, the parameter space bounds may not contain the model trained on the clean dataset. Therefore, our certificates are only valid if the adversary strength does not exceed the specification used.
> > >
> > > *Data ordering assumption:*
> > > The worst case poisoning attack indeed probably does not place all $n$ points in the same batch. In our response above, we referred to the “worst-case” with respect to our guarantees. For example, given that we certify $n$ poisoned points per batch, an adversary capable of poisoning $n+1$ points could bypass our certificates by placing all $n+1$ points in a single batch, which we provide no certificate against. This may or may not be the most effective attack strategy, but regardless we cannot provide dataset-level certificates against such an adversary capable of poisoning $n$ points unless we consider the “worst-case” of $n$ points per batch.
> > >
> > > *Scalability:*
> > > Our experiments demonstrate the scaling of our certificates with respect to both the model depth and model width (Figure 2), which both vary the number of model parameters. We do not explicitly study the number of model parameters vs tightness, and it is unclear how to do this in general (increasing width/depth affect AGT differently). Moreover, as the number of input features increases (and correspondingly, the number of parameters in the first layer), the tightness of our certificates will decrease. Since each parameter admits its own interval bound, increasing the number of parameters introduces additional over-approximation via the various mechanisms discussed in the paper (e.g., the element-wise relaxation of Theorem 3.4), and in works applying similar bounding techniques in adversarial verification.

---

> > > > ### Comment · Reviewer_YK3j · 2025-07-27
> > > >
> > > > Thanks for your clarifications!
> > > >
> > > > clean dataset assumption. Based on my understanding, when there are $k$ points already poisoned in the dataset, the algorithm may or may not select these points in line 8 Algorithm 1. Therefore, while intended certificate for $n$ poisoned samples, it may compute the bounds for at most $n+k$ samples (while these additional samples $k$ samples are not given the strongest adversaries capabilities). This bound is looser than desired $n$ samples while I understand this could still be considered as a reasonable (but looser) bound for the desired $n$ samples, and I think this discussion could be added to the manuscript.
> > > >
> > > > data ordering. I think it would be helpful to state that the analysis in the manuscript is based on per-batch data, perhaps in the problem setup section.

---

> > > > > ### Author Response · Authors · 2025-07-28
> > > > > **Response to Reviewer YK3j**
> > > > >
> > > > > Thanks for your response. We will add clarifying comments to the manuscript to better explain the per-batch analysis.
> > > > >
> > > > > Regarding your example stated above, our bounds (which compose the worst-case $n$ poisoned points per batch) are indeed loose with respect to a dataset-level adversary capable of poisoning $n$ points (which results in our bounds being valid for $n+k$ points, as long as no batch exceeds $n$ poisoned datapoints). In fact, our algorithm returns a certificate for $Bn$ points being poisoned (where $B$ is the number of batches in the dataset), assuming that the poisoned data are distributed between training batches such that each batch has at most $n$ poisoned points. There are batch-level threat models for which this assumption could apply more exactly (e.g., as in the dynamic attack model of [1]).
> > > > >
> > > > > [1] Bose, Avinandan, et al. "Keeping up with dynamic attackers: Certifying robustness to adaptive online data poisoning." arXiv preprint arXiv:2502.16737 (2025).

---

### Review · Reviewer_CbJE · 2025-06-13

**Summary Of Contributions:**

This paper proposes a method to certify the robustness of neural networks against data poisoning attacks, which perturb the feature or label of training data. In particular, the algorithm returns the upper bound of the worst-case adversary, such as the lowest accuracy or the highest MSE that the attack could achieve. Intuitively, this method leverages convex relaxation to bound the reachable set of all possible parameters that could be obtained by the training procedure and theoretically translates that bound to the adversary's objective. The experiments demonstrate the certified robustness on simple classification and regression problems.

**Audience:**

Yes

**Claims And Evidence:**

Yes

**Requested Changes:**

This work could be significantly improved if the experiments on large-scale datasets and large model are provided.

**Strengths And Weaknesses:**

Strengths:
- This paper studies a novel problem, that is to provide certified robustness against both bounded and unbounded data poisoning attacks.
- The idea of using the reachable set is interesting.
- The paper discusses many aspects of the algorithm, such as complexity, certifying a single query, and combining forward and backward pass.

Weaknesses:
- This algorithm has high complexity to have tight bound ($O(bm^2n^3)), which may not be practical for real-world applications. The experiments are only conducted on small-scale datasets and models.
- Fig. 3 shows that the certified robustness is not quite high, and improving that trades off the nominal accuracy.
- Fig. 4 implies that randomized smoothing consistently outperforms the proposed method when $n>50$, which is not unrealistic.

---

> ### Author Response · Authors · 2025-07-01
> **Response to Reviewer CbJE**
>
> We thank the reviewer for their feedback on our submission. The primary concern raised relates to the scalability and practicality of our method when applied to larger models and datasets. As noted in our response to Reviewer 4DXp, this work introduces a novel and general framework that extends  inference-time verification techniques to training-time certification. Our framework is flexible and can incorporate tighter methods developed in the verification literature, such as optimization-based approaches, which may improve scalability. Additionally, integrating our framework with orthogonal developments such as ensemble methods for certified robustness presents another promising direction for scaling to larger settings. While we expect future work to advance and refine the techniques introduced here, we believe this paper establishes an important initial foundation for further research in this new area.

---

### Author Response · Authors · 2025-07-01
**Global Response to Reviewers**

We thank all reviewers for their thoughtful feedback and have addressed each of the specific concerns in our individual responses. In addition, we have revised our manuscript with the following updates, which are highlighted in blue in the revised PDF:

- Added new experimental results comparing the tightness of IBP and CROWN bounds.
- Clarified notation and terminology used in the theoretical sections.
- Moved the descriptions of bound propagation and interval arithmetic from the appendix into the main text to enhance readability and improve the overall flow.

---

### Decision · Action_Editor_PqUv · 2025-08-04

**Recommendation:** Accept as is

**Audience:**

Yes

**Audience Explanation:**

Data poisoning is an important unsolved problem in ML, and most existing mitigations are empirical. This paper's novel framework of certifying robustness theoretically is interesting and is appreciated by all reviewers. I believe the TMLR audience will be interested in findings of this paper.

**Claims And Evidence:**

Yes

**Claims Explanation:**

The paper proposes a framework for certifying robustness to data poisoning attacks. The authors prove theoretically that their robustness bound holds in a computable parameter region, and empirically validate their result on linear models and small neural networks.

Reviewers generally find the approach to be novel and interesting, but have concerns about its practicality and scalability to large neural networks. Reviewer YK3j also points out two potentially misleading claims about the framework's assumptions, which have been addressed in the revision.

The authors provide clear description and analysis of their method, along with empirical result to back up their claims. Thus, I believe the paper meets this criteria despite its limitations in practicality and scalability.